# Genome-wide DNA methylation and gene expression in human placentas derived from assisted reproductive technology

Pauliina Auvinen[1], Jussi Vehviläinen[1], Karita Rämö[1], Ida Laukkanen[1], Heidi Marjonen-Lindblad[1], Essi Wallén[1], Viveca Söderström-Anttila[2], Hanna Kahila[3], Christel Hydén-Granskog[3], Timo Tuuri[3], Aila Tiitinen[3] & Nina Kaminen-Ahola [1] ✉

## Abstract

**Background** Assisted reproductive technology (ART) has been associated with increased risks for growth disturbance, disrupted imprinting as well as cardiovascular and metabolic disorders. However, the molecular mechanisms and whether they are a result of the ART procedures or the underlying subfertility are unknown.

**Methods** We performed genome-wide DNA methylation (EPIC Illumina microarrays) and gene expression (mRNA sequencing) analyses for a total of 80 ART and 77 control placentas. The separate analyses for placentas from different ART procedures and sexes were performed. To separate the effects of ART procedures and subfertility, 11 placentas from natural conception of subfertile couples and 12 from intrauterine insemination treatments were included.

**Results** Here we show that ART-associated changes in the placenta enriche in the pathways of hormonal regulation, insulin secretion, neuronal development, and vascularization. Observed decreased number of stromal cells as well as downregulated *TRIM28* and *NOTCH3* expressions in ART placentas indicate impaired angiogenesis and growth. DNA methylation changes in the imprinted regions and downregulation of *TRIM28* suggest defective stabilization of the imprinting. Furthermore, downregulated expression of imprinted endocrine signaling molecule *DLK1* associates with both ART and subfertility.

**Conclusions** Decreased expressions of *TRIM28*, *NOTCH3*, and *DLK1* bring forth potential mechanisms for several phenotypic features associated with ART. Our results support previous procedure specific findings: the changes associated with growth and metabolism link more prominently to the fresh embryo transfer with smaller placentas and newborns, than to the frozen embryo transfer with larger placentas and newborns. Furthermore, since the observed changes associate also with subfertility, they offer a precious insight to the molecular background of infertility.

## Plain language summary

For those that struggle with conception, medical and scientific methods called Assisted Reproductive Technology (ART) may help. However, ART have been associated with increased risks for negative medical outcomes for babies. Whether these risks are caused by ART use or the underlying condition of subfertility (less than ideal natural conception outcomes) are not known. Here we looked at the effects of ART and subfertility by studying specific genetics in placenta and newborn's characteristics. We show that changes in genetics in the placenta from ART use are linked to hormonal control, insulin secretion, and brain and blood vessel development. Although the observed changes are subtle, they can contribute to risks for metabolic and heart disorders as well as growth disturbances in newborns. Our results provide important evidence for the effect of medical outcomes associated with both ART and subfertility.

Approximately one in six couples experience infertility worldwide and the usage of Assisted Reproductive Technology (ART) has increased over the last decades. To date over ten million children have been conceived through these technologies[1]. The main procedure of ART is in vitro fertilization (IVF), which may also include intracytoplasmic sperm injection (ICSI), in which a single sperm cell is injected into the oocyte cytoplasm. In addition to the freshly transferred embryos (FRESH), also embryo cryopreservation and frozen embryo transfer (FET) can be a part of IVF or ICSI procedures. Less invasive in vivo fertility treatments, such as intrauterine insemination (IUI) and hormone treatments to induce ovulation, are also widely used.

[1]Environmental Epigenetics Laboratory, Department of Medical and Clinical Genetics, Medicum, University of Helsinki, Helsinki, Finland. [2]The Family Federation of Finland, Fertility Clinic and University of Helsinki, Helsinki, Finland. [3]Department of Obstetrics and Gynecology, Helsinki University Hospital and University of Helsinki, Helsinki, Finland. ✉e-mail: nina.kaminen@helsinki.fi

Although children conceived by using the treatments are generally healthy, ART has been associated with increased risks for adverse obstetric and perinatal outcomes such as preterm birth (PTB), low birth weight (LBW), birth defects, and placental anomalies[2,3]. Additionally, long-term effects such as rapid postnatal growth, female early onset puberty as well as cardiovascular and metabolic disorders have been associated with the ART phenotype[4–6]. An increased risk for certain imprinting disorders[7,8] and a tendency towards higher risk of neurodevelopmental disorders has furthermore been reported[9]. There are method-associated risks, and it has been reported that FRESH derived children have a higher risk for PTB, LBW, and for being small for gestational age (SGA) compared to natural conception. A higher risk of pre-eclampsia and a higher risk for being large for gestational age (LGA) has been noted in pregnancies after FET compared to FRESH[10]. However, it is unclear whether these associations are the result of ART technology *per se* or the underlying subfertility.

The mechanisms by which ART or infertility increases the risk for adverse outcomes are unknown. ART has been suggested to affect fetal development through epigenetic modifications as the procedures take place during extensive epigenetic reprogramming in the gametogenesis and early embryogenesis. Indeed, ART-associated DNA methylation (DNAm) profiles have been reported in several epigenome-wide association studies (EWASs) of human blood[11–18] and placenta[19–22]. Increasing evidence suggest that the placenta is particularly susceptible to epigenetic changes caused by ART and/or infertility[23–25] and alterations have been found especially in epigenetically controlled repetitive elements (REs)[26–28] and parent-of-origin monoallelically expressed imprinted genes[19–22]. However, despite of some repetitive candidate loci, particularly related to imprinted genes, the ART-associated DNAm alterations are mainly inconsistent[29,30]. Furthermore, ART procedure-specific EWASs as well as genome-wide gene expression studies for placentas are scarce and studies controlled for placental cell type heterogeneity are lacking.

Here, to gain a deeper understanding of the ART- and subfertility-associated molecular changes and phenotypes, we explored ART-associated genome-wide DNAm by microarrays (Illumina's Infinium MethylationEPIC) and gene expression by 3'mRNA sequencing (mRNA-seq) of full-term placental samples from ART and naturally conceived singleton newborns. The effect of cell type heterogeneity was excluded by adjusting the DNAm analyses by placental cell types. Furthermore, ART-associated phenotypic features were examined. The procedure-specific changes were investigated by analyzing the ART samples in subgroups: newborns derived from IVF or ICSI methods as well as from FRESH or FET procedures. Moreover, to separate the effects of decreased fertility from ART procedures, we analyzed placental samples from pregnancies of couples who had went through IUI as well as from subfertile (SF) couples who were about to start or already started ART treatment but got pregnant spontaneously.

In addition to ART procedure-specific analyses, we performed sex-specific analyses by examining placentas of male and female newborns separately, including sex chromosomes. Males and female fetuses have been shown to differ in terms of placental DNAm[31,32] as well as in their ability to adapt to adverse intrauterine environments[33,34]. In the ART studies, it has been shown that males have a higher risk to be LGA compared to females[35–37], but still only a few sex-specific EWASs have been performed so far[12,18,22].

Here, we bring forth three candidate genes, *tripartite motif-containing protein 28* (*TRIM28*), *notch receptor 3* (*NOTCH3*), and *delta like non-canonical Notch ligand 1* (*DLK1*), which together with the observed placental molecular changes elucidate the mechanisms underlying ART-associated phenotypes. The downregulation of *TRIM28*, a stabilizer of imprinting, and DNAm changes in imprinted genes could explain ART-associated increased risk for imprinting defects. Furthermore, our findings clarify the contribution of different ART procedures, sex of the child, and subfertility to the molecular alterations.

## Methods

### Epigenetics of ART (epiART) cohort and ethics statement

Finnish, white couples applied to fertilization treatment in the Fertility clinic of the Family Federation of Finland or the Reproductive Medicine Unit of Helsinki University Central Hospital, Finland were recruited to this study and the samples were collected during the years 2015–2021. The epiART cohort includes 80 newborns fertilized using ART (IVF $n = 50$ and ICSI $n = 30$) of which 42 using FRESH and 38 using FET embryos. Moreover, the cohort includes newborns from pregnancies of couples who had went through IUI as well as from SF couples who were about to start or already started ART treatment but got pregnant spontaneously. More detailed information about the number of samples, newborns´ phenotypes and maternal characteristics is presented in Table 1 and Supplementary Data 1. Parental infertility diagnoses of ART, IUI, and SF samples were categorized into male or female factor infertilities, infertility caused by multiple causes, and unexplained infertility (Supplementary Fig. 1 and Supplementary Data 2). Four cases in the ART group (5%) (IVF 4% and ICSI 3.3%) and one in the IUI group (8.3%) used medication for chronic somatic disease hypothyroidism. Two ART cases (2.5%), both in IVF group (4%), had a psychiatric disorder diagnosis and the other one used medication for it (anxiety disorder). One in the IUI group (8.3%) had medication for a bipolar disorder. Pregnancy related disorders such as gestational diabetes mellitus occurred in seven ART cases (8.8%) (IVF 12%, and ICSI 33.3%), two IUI (16.7%) and two SF (18.2%), hepatogestosis in three ART cases (3.8%) (IVF 4%, ICSI 3.3%) and pregnancy related hypertension in one IUI (8.3%) and one SF (9.1%). A total of 16 ART (20%) (IVF 22% and ICSI 17%), four IUI (33%), and three SF (27%) deliveries were cesarean sections (CSs). The naturally conceived controls ($n = 77$) were full term neonates born from uncomplicated pregnancies of healthy Finnish, white mothers collected during the years 2013–2015 in Helsinki University Hospital, Finland. Five (6.5%) of the deliveries were CSs. Informed parental consents for the use of samples and data related to newborns and parents themselves were obtained from all participants and the study was approved by the Ethics Committee of Helsinki University Central Hospital (386/13/03/03/2012, 285/13/03/03/2013).

Differences in newborn birth measures and maternal characteristics between study groups were calculated, depending on the data distribution, by parametric two-tailed Student's *t*-test or nonparametric Mann-Whitney *U* test in the case of two groups, and by parametric One-Way ANOVA followed by Bonferroni *post hoc* or nonparametric Kruskal-Wallis test followed by pairwise comparisons in the case of multiple

**Table 1 | Samples sizes of each study group in phenotype, genome-wide DNAm, and mRNA-seq analyses**

| Group | Phenotypes and Genome-wide DNAm | Genome-wide mRNA-seq |
|---|---|---|
| | N (male/female) | N (male/female) |
| **Control** | 77 (42/35) | 39 (19/20) |
| **ART** | 80 (36/44) | 59 (29/30) |
| IVF | 50 (24/26) | 38 (20/18) |
| IVF-FRESH | 25 *(10/15)* | 19 *(8/11)* |
| IVF-FET | 25 *(14/11)* | 19 *(12/7)* |
| ICSI | 30 (12/18) | 21 (9/12) |
| *ICSI-FRESH* | *17 (4/13)* | *11 (3/8)* |
| *ICSI-FET* | *13 (8/5)* | *10 (6/4)* |
| FRESH | 42 (14/28) | 30 (11/19) |
| FET | 38 (22/16) | 29 (18/11) |
| **IUI** | *12 (8/4)* | *10 (6/4)* |
| **SF** | *11 (5/6)* | *10 (4/6)* |

The main study groups (bolded) and subgroups (unbolded) are presented with sample sizes in each analysis including the number of samples in the sex-specific analyses. The subgroups or sex-specific samples which are not included in the analyses are written in italics.
*ART* assisted reproductive technology, *IVF* in vitro fertilization, *ICSI* intracytoplasmic sperm injection, *FRESH* fresh embryo transfer, *FET* frozen embryo transfer, *IUI* insemination, *SF* subfertile.

groups. Differences in birth weight (BW), birth length (BL), and head circumference (HC) of the newborns were calculated using both anthropometric measures and the SDs of measures based on Finnish growth charts[38], in which gestational age (GA) at birth, twinning, parity, and gender were considered when calculating the SDs (z-scores) of birth measures. SGA and LGA are characterized by weight and/or length at least 2 SD scores below or above the mean.

## Sample collection and preparation

Placental samples of newborns were collected immediately after delivery. When this was not possible, the placenta was stored in the fridge for a maximum of 12 h and only DNA was extracted for further analyses. The placental biopsies (1 cm³) were collected from the fetal side of the placenta within a radius of 2–4 cm from the umbilical cord, rinsed in cold 1× PBS, fixed in RNAlater® (Thermo Fisher Scientific), and stored at −80 °C. Placental genomic DNA was extracted from one to four (3.5 on average) pieces of placental tissue samples using commercial QIAamp Fast DNA Tissue or AllPrep DNA/RNA/miRNA Universal Kits (Qiagen) or standard phenol-chloroform protocol. RNA was extracted from the same placental pieces as DNA (2.8 pieces on average) by AllPrep DNA/RNA/miRNA Universal Kit. RNA quality was assessed using an Agilent 2100 Bioanalyzer (Agilent Technologies, Inc.), which was provided by the Biomedicum Functional Genomics Unit (FuGU) at the Helsinki Institute of Life Science and Biocenter Finland at the University of Helsinki.

## DNAm microarrays

**DNAm measurements and data processing.** Genomic DNA (1000 ng) from available placental samples (sample information in Supplementary Data 1) was sodium bisulfite-converted using the Zymo EZ DNAm™ kit (Zymo Research), and genome-wide DNAm was assessed with Infinium Methylation EPIC BeadChip Kit (Illumina) following a standard protocol at the Institute for Molecular Medicine Finland. The raw DNAm dataset was pre-processed, quality controlled, and filtered using ChAMP R package[39] with default settings: the detection P-value cutoff was set at $P = 0.01$, and probe bead count >3 in at least 95% of samples. Although all samples passed these quality control thresholds, 45,227 probes were excluded from the subsequent steps. Type-I and Type-II probes were normalized using the *adjustedDasen* method[40] in wateRmelon R package[41]. Potential effects caused by technical factors and biological covariates were studied from singular value decomposition (SVD) plots. The data was corrected for the effects of array, slide, batch, DNA extraction method, and placental type (stored in a fridge before sampling or not) by the Empirical Bayes method using the R package ComBat[42]. After ComBat adjustment, probes located in sex chromosomes and probes binding to polymorphic and off-target sites[43] were filtered. Moreover, probes in Finnish-specific single nucleotide polymorphisms (SNPs), which overlap with any known SNPs with global minor allele frequency were removed. Population-specific masking information was obtained from Zhou et al. [44].

Subsequently, a total of 709,230 probes were retained for further downstream analysis. Probes in sex chromosomes (15,741 probes for males and 15,723 probes for females) were included in sex-specific analyses. Annotation information was merged to corresponding probes from IlluminaHumanMethylationEPICanno.ilm10b4.hg19 R package[45], which is based on the file "MethylationEPIC_v-1-0_B4.csv" from Illumina[46]. Probe genomic locations relative to gene and cytosine-phosphate-guanine (CpG) island were annotated based on University of California, Santa Cruz (UCSC) database. If the location information was missing, probe was marked as "unknown." In the case of multiple location entries, group "others" was used. Otherwise, the following abbreviations were used: TSS1500: 1500 bp upstream of transcription start site (TSS), TSS200: 200 bp upstream of TSS, UTR: untranslated region, N_shelf: north shelf, N_shore: north shore, S_shore: south shore, S_shelf: south shelf. Probes were annotated to genes based on, in addition to UCSC, also GENCODE Basic V12 and Comprehensive V12 databases.

## Differentially methylated position analysis

Genome-wide differential DNAm analysis by using M values was performed by R package Limma[47], and the linear model was adjusted based on SVD plots for batch as well as biological covariates newborn sex, maternal age, and maternal pre-pregnancy body mass index (BMI). Planet R package[48] was used to count placental cell-type fractions by CIBERSORT method with unfiltered data and used as an adjusting factor in the model. For male samples, the model was adjusted for cell type, DNA extraction method, parity, and BMI, and for female samples, the model was adjusted for cell type. β values were used for visualization and interpretation of the results. Due to the observed inflation, CpGs were considered as significant (hereafter differentially methylated positions, DMPs) when DNAm difference in the effect size was ≥5% ($\Delta\beta \leq -0.05$ and $\Delta\beta \geq 0.05$) and FDR-corrected P-value smaller than 0.05. Benjamini-Hochberg procedure was used to control for FDR. To calculate the differences in placental cell type composition between the study groups, depending on the data distribution, parametric or nonparametric two-tailed t-test in the case of two groups, and parametric One-Way ANOVA followed by Bonferroni *post hoc* or nonparametric Kruskal-Wallis test followed by Wilcoxon Rank Sum Exact test in the case of multiple groups were used.

**Differentially methylated region and pathway analyses.** DMRcate R package[49] was used to analyze differentially methylated regions (DMRs). The method uses minimum description length for detecting region boundaries in DMR identification. DMRcate was adjusted to determine probes (≥3) in a region with maximal allowed genomic distance of 1000 bp having FDR < 0.05. Pcutoff parameter was set to 0.05. DMRs with Fisher's combined probability test P < 0.05 were considered significant. DMRs were annotated to imprinted genes based on Geneimprint database[50]. Pathway analysis was performed for significant DMRs by *goregion* function in missMethyl R package[51]. Gene Ontology (GO) knowledgebase for identifying significantly enriched biological process (BP) terms and KEGG knowledgebase were used as a source. When the GO terms were not significant after FDR correction, the terms with the nominal P-value < 0.05 were reported.

**Genome-wide average methylation analysis.** β values of all probes in the array (normalized, ComBat adjusted, and filtered data) were used to calculate placental genome-wide average methylation (GWAM)[52] levels sample-wise. Differences between the study groups were calculated by parametric or nonparametric two-tailed t-test depending on the data distribution in the case of two groups. In the case of multiple groups, One-Way ANOVA followed by Tukey's honestly significant difference (HSD) test was used.

**RE DNAm analysis.** Processed DNAm data (M values) was used to predict DNAm in Alu, endogenous retrovirus (ERV), and long interspersed nuclear element 1 (LINE1) elements sample-wise using Random Forest-based algorithm implemented by REMP R package[53]. Less reliable predicted results were trimmed according to quality score threshold 1.7 and missing rate 0.2 (20%). Differences between the study groups were calculated by parametric or nonparametric two-tailed t-test depending on the data distribution in the case of two groups. In the case of multiple groups, One-Way ANOVA followed by Tukey's HSD test was used.

**Imprinting control region analysis.** In total of 826 probes located at imprinting control regions (ICRs) based on Ochoa et al. [54] were compered between ART and control placentas by two-tailed Student's t-test. Bonferroni adjustment was used for multiple comparison.

## Traditional bisulfite sequencing

To examine the DNAm profiles at the *DLK1- iodothyronine deiodinase 3* (*DIO3*) ICR, a total of eight ART (four IVF, four ICSI, two/sex), four SF (two/sex), and six control placental samples (three/sex) were subjected to bisulfite sequencing. Samples were chosen based on mRNA-seq counts, control

samples having the highest and ART and SF samples the lowest counts. Paternal and maternal alleles were distinguished according to rs1884539(A/G) and rs75998174(A/G) polymorphisms at the ICR. Primers were designed to incorporate the polymorphisms using Bisulfite Primer Seeker tool (Zymo Research)[55] (forward 5'GGTTTATAGTTGTTTATGGTTTGTTAAT and reverse 5'CTCCAACAAAAATTCCTTAAACTAAATT). The 451 bp amplicon (chr14:101,277,375-101,277,826) covered 12 CpG sites at the ICR. Two separate bisulfite conversions were performed for 500 ng of genomic DNA using EZ DNA Methylation™ kit (Zymo Research) and pooled afterwards. To avoid possible PCR bias, three independent 20 μl PCR reactions (HotStar PCR kit, Qiagen) were performed per sample using annealing temperature 56 °C. PCR reactions were gel isolated, and the three reactions of each sample were pooled and purified using NucleoSpin Gel and PCR Clean-up Kit (Macherey-Nagel). The purified PCR fragments were ligated into pGEM®-T Easy Vector (Promega) and cloned by standard protocol. The recombinant-DNA clones were purified using NucleoSpin® Plasmid Easy-Pure kit (Macherey-Nagel) according to manufacturer's instructions. The sequences were analyzed by BIQ Analyzer[56] excluding the clones with lower than 90% conversion rate from the dataset. A total of eight to 30 clones of each individual were successfully sequenced (Supplementary Fig. 2). DNAm differences in CpG sites between study groups (control, ART, and SF) were calculated by One-Way ANOVA followed by Bonferroni *post hoc*.

### mRNA-seq analysis

**Differential expression analysis.** Drop-seq pipeline[57] was used to construct the mRNA-seq count table for available placental RNA samples (sample information in Supplementary Data 3) provided by FuGU. A total of 34,989 transcripts were identified for downstream analysis. Principal component analysis implemented in DESeq2[58] was used to identify batch effects, and ComBat-seq[59] was used to adjust separate mRNA-seq batches. Differential expression analysis was performed by DESeq2 R package, with model adjusting for newborn sex. Genes were considered significantly differentially expressed when FDR-corrected *P*-value was <0.05. Benjamini-Hochberg procedure was used to control for FDR.

*Pathway analysis. enrichgo* function in R package clusterProfiler version 4.0[60] was used to perform gene-set enrichment analysis for significant differentially expressed genes (DEGs). The GO knowledgebase was used as the source for identifying significantly enriched BP terms (FDR-corrected *q*-value < 0.05). Benjamini-Hochberg procedure was used to control for FDR.

### Correlation analysis

The within sample gene expression correlations were calculated using ComBat-seq-adjusted and Transcripts Per Million-normalized mRNA-seq counts and the correlations between gene expression, newborn phenotype, and maternal characteristics using ComBat-seq-adjusted and DESeq2-normalized mRNA-seq counts by Spearman rank correlation.

### Statistics and reproducibility

All statistical analyses were conducted in R versions 4.2.3, 4.3.1, and 4.3.3[61], IBM SPSS Statistics for Windows, version 29.0 (IBM Corp.), or GraphPad Prism 9 software (GraphPad Software, Inc.). All data are expressed as the mean with ±SD for a normal distribution of variables. Statistical analyses were performed as described in the relevant method sections and in the figure legends. Effect sizes were estimated using Cohen's *d* for parametric tests and Rank-Biserial Correlation for nonparametric tests. The number of biologically independent samples used in genome-wide DNAm and mRNA-seq analyses as well as in phenotype analysis are presented in Table 1 as well as in Supplementary Data 1 and 3. Traditional bisulfite sequencing was performed for a total of eight ART (four IVF, four ICSI, two/sex), four SF (two/sex), and six control biologically independent samples (three/sex).

### Reporting summary

Further information on research design is available in the Nature Portfolio Reporting Summary linked to this article.

## Results

### Participant characteristics and placental phenotypes

General characteristics of 80 ART and 77 naturally conceived control newborns as well as their mothers were compared. All ART and subgroup-specific information of the phenotype analysis can be found in the Supplementary Data 1. In general, the phenotypes did not differ significantly between ART and control newborns. A total of five SGA (6.3%) (IVF male, two IVF females and two ICSI females, all FRESH) and five LGA (6.3%) (IVF-FRESH, ICSI-FRESH, and ICSI-FET males as well as ICSI-FRESH and ICSI-FET females) newborns were in the ART group, and two SGA newborns (2.6%, male and female) in controls. The GA of ART newborns was significantly shorter, particularly in females (ART: *n* = 44, control: *n* = 35) (*r* = −0.235, *P* = 0.003 and *r* = −0.326, *P* = 0.004, respectively, Mann-Whitney U), and this was driven by IVF and FRESH newborns (*n* = 50, *n* = 42, respectively). In the ART group, one (1.3%) IVF newborn was preterm (gestational week 36 + 5). The mothers of ART newborns were significantly older, and they gained less weight during pregnancy compared to the mothers of control newborns (*d* = 0.931, *P* < 0.0001, Student's *t*-test, *r* = −0.249, *P* = 0.002, Mann-Whitney U, respectively).

Furthermore, IUI (*n* = 12) and SF (*n* = 11) newborns as well as their mothers were compared (Supplementary Data 1). The GA of SF newborns was significantly shorter compared to controls (*r* = −0.296, *P* = 0.005, Mann-Whitney U) and one female LGA newborn was observed in this group. The mothers of IUI and SF newborns were significantly older, and the BMI of the IUI mothers was higher compared to controls (*d* = 0.765, *P* = 0.016, Student's *t*-test, *d* = 0.783, *P* = 0.017, Student's *t*-test, *r* = 0.291, *P* = 0.006, Mann-Whitney U, respectively).

When the placentas of ART and control newborns were compared, no significant difference in weights was observed (Supplementary Data 1). However, in ART subgroups, FET placentas (*n* = 38) were significantly heavier compared to FRESH (*n* = 42) (*d* = 0.586, *P* = 0.022, One-way ANOVA followed by Bonferroni post hoc). The DNAm profiles of placental samples and cell type fragment analysis were utilized to determine potential differences between ART and control placentas in five major placental cell types: syncytiotrophoblasts, trophoblasts, stromal cells, Hofbauer cells, and endothelial cells (Supplementary Fig. 3). When comparing the fragments between ART (*n* = 80) and control (*n* = 77) samples, we observed significantly higher number of trophoblasts in the ART placentas (*d* = 0.432, *P* = 0.04, One-way ANOVA followed by Tukey's HSD), which was driven by males (*d* = 0.467, *P* = 0.041, Student's *t*-test) (Supplementary Fig. 4). Contrary, significantly lower number of stromal cells was observed in ART placentas (*d* = −0.535, *P* = 0.011, One-way ANOVA followed by Tukey's HSD) as well as IVF (*n* = 50, *r* = −0.326, *P* = 0.001, Kruskal-Wallis test followed by Wilcoxon Rank Sum test), FRESH (*n* = 42, *d* = −0.562, *P* = 0.011, One-way ANOVA followed by Tukey's HSD), and FET (*n* = 38, *d* = −0.501, *P* = 0.033, One-way ANOVA followed by Tukey's HSD) subgroup placentas compared to controls. According to sex-specific analyses, there were significantly lower number of stromal cells in female ART (*n* = 44, *d* = −0.711, *P* = 0.003, Student's *t*-test), IVF (*n* = 26, *d* = −0.914, *P* = 0.004, One-way ANOVA followed by Tukey's HSD), and FRESH (*n* = 28, *d* = −0.796, *P* = 0.009, One-way ANOVA followed by Tukey's HSD) placentas compared to control females (*n* = 35) (Supplementary Fig. 4). Regarding endothelial cells, the number was significantly higher in FRESH placentas compared to FET (*d* = 0.627, *P* = 0.02, One-way ANOVA followed by Tukey's HSD). Furthermore, to exclude the variation caused by different ART procedures, we compared FRESH and FET placentas derived only from the IVF to the controls. Significantly more endothelial cells were also observed in the IVF-FRESH placentas (*n* = 25) compared to the IVF-FET placentas (*n* = 25) (*d* = 0.926, *P* = 0.004, One-way ANOVA followed by Tukey's HSD).

### ART-associated genome-wide DNAm in placenta

To investigate GWAM level in the ART (*n* = 80) and control (*n* = 77) placentas, we used the DNAm level of all 709,230 probes in the microarrays (general characteristics in Supplementary Data 1). Furthermore, we

examined ART-associated DNAm in REs by comparing the predicted mean DNAm level of CpGs in Alu, ERV, and LINE1 sequences in ART and control placentas. However, we did not observe significant ART-associated alterations in GWAM levels either overall or at any genomic locations relative to gene or CpG island, or in REs (Supplementary Fig. 5 and 6, respectively).

ART-associated genome-wide DNAm was explored by using linear regression model adjusted for batch, cell type, sex, as well as maternal pre-pregnancy BMI and age as covariates. The analysis resulted in 6814 significantly differentially methylated CpG sites (3739 hypo- and 3075 hypermethylated) with FDR < 0.05 (Fig. 1a, Supplementary Data 4). To separate the most prominent changes and to minimize false positive hits due to the observed inflation, we focused on the CpG sites with DNAm difference in the effect size of ≥5% between ART and control placentas, which are termed as differentially methylated positions (DMPs). There were 260 DMPs associating with 185 genes in ART placentas (FDR < 0.05, $\Delta\beta \leq -0.05$ and $\Delta\beta \geq 0.05$), of which 129 were hypo- and 131 hypermethylated. Further, we tested for DMRs defined as a region with maximal allowed genomic distance of 1000 bp containing three or more CpGs with $P < 0.05$ according to Fisher's combined probability test. A total of 822 DMRs associating with 818 genes in ART placentas were revealed (Supplementary Data 5).

To get a comprehensive picture on the BPs in which ART-associated DMRs cluster, we performed pathway analyses (Supplementary Data 6). GO enrichment analysis of ART-associated DMRs revealed BPs involved in cell-cell adhesion via plasma-membrane adhesion molecules (FDR < 0.05), nervous system development, hormone secretion and transport, as well as regulation of cellular response to growth factor stimulus ($P < 0.05$). According to KEGG enrichment analysis, ART-associated DMRs linked to terms such as gonadotropin-releasing hormone (GnRH) secretion, endocrine resistance, tumor necrosis factor (TNF) signaling pathway, growth hormone synthesis, secretion, and action, as well as estrogen signaling pathway ($P < 0.05$) (Fig. 1c).

Furthermore, placental DMRs in the current study associated with *CHRNE, NECAB3, FSCN2, PDE11A, MRPS22, PRR23A, SLC9B1, SLC9B2, RGR, GALNT9, RPH3AL, SFT2D3,* and *WDR33* as well as both DMRs and DMPs with *PCDHGB4, APC2, MSX1, RASL11B, CYP2E1,* and *SPNR,* which all had DNAm changes in the same direction as in the blood samples of ART derived individuals in previous EWASs[14,16,17]. Moreover, similar changes in *RIMS1, FDFT1, FGF5, MYO7A, TRIM72, C4orf51, KCNIP1, FRZB,* and *NBR1* have been observed in previous studies, in which first-trimester or full-term ART placentas were compared to placentas from IUI and SF pregnancies[20,21]. Also, *ALDH4A1, RASGEF1A, GRIN2C, KCNIP2, KIFC2, FOXH1, MUC5B,* and *CELF5* with DNAm changes in the same direction have been previously associated with IVF in first-trimester placentas[62].

## ART-associated genome-wide gene expression in placenta

To study genome-wide ART-associated gene expression, we performed mRNA-seq for ART ($n = 59$) and control placentas ($n = 39$) (general characteristics in Supplementary Data 3). When the mRNA-seq model was adjusted by sex, we observed 71 significant DEGs (FDR < 0.05) of which 53 were downregulated and 18 upregulated (Fig. 1b, Supplementary Data 7). One of the most significantly downregulated gene was *DLK1* (Fig. 1d, Supplementary Data 8). This paternally expressed imprinted gene is essential for embryonic development and growth, and its malfunction has been connected to metabolic abnormalities, such as obesity, Type II diabetes mellitus (T2D), and hyperlipidemia in previous human studies[63]. Other interesting downregulated DEGs were *TRIM28* and *NOTCH3* (Fig. 1d, Supplementary Data 8). *TRIM28,* is a mediator of epigenetic modifications and essential in decidualization and implantation[64]. *NOTCH3,* which is expressed mainly in vascular smooth muscle and pericytes, is needed for the development of fully functional arteries in mouse[65] and to control human trophoblast stem cell expansion and proliferation[66]. Interestingly, also ART-associated DMRs were observed in its family member *NOTCH1,* which mediates uterine stromal

differentiation and is essential for decidualization and implantation in mouse[67].

When all the placentas were analyzed by Spearman rank correlation, moderate correlations were seen between *TRIM28* and *DLK1* ($r = 0.357$, $P < 0.0001$, $n = 118$) as well as between *TRIM28* and *NOTCH3* ($r = 0.358$, $P < 0.0001$, $n = 118$) expressions (Supplementary Data 9). However, when the control and ART placentas were analyzed separately, there was no correlation between *TRIM28* and *DLK1,* and the correlation between *TRIM28* and *NOTCH3* was only seen in the control ($r = 0.472$, $P = 0.002$, $n = 39$) but not in the ART ($n = 59$) placentas. Interestingly, although there is no evidence of direct interaction between *DLK1* and *NOTCH3,* correlations between the expressions of the genes in all ($r = 0.518$, $P < 0.0001$, $n = 118$), control ($r = 0.419$, $P = 0.008$, $n = 39$), ART ($r = 0.427$, $P = 0.001$, $n = 59$), and SF ($r = 0.733$, $P = 0.016$, $n = 10$) placentas were observed.

According to the GO:BP enrichment analysis, ART-associated gene expression is linked predominantly to vasculogenesis including endothelial cell development (*PECAM1, S1PR3, COL18A,* and *ROBO4*), renal vasculature development (*PECAM1, NOTCH3,* and *CD34*), and foam cell differentiation (*AGTR1, ABCA1,* and *CETP*) (FDR-corrected $q$-value < 0.05) (Fig. 1e, Supplementary Data 10). Interestingly, foam cells are a sign for vascular changes in the placenta, the so-called acute atherosis, which has been observed frequently in non-transformed spiral arteries in pregnancies associated with pre-eclampsia, SGA, fetal death, spontaneous PTB, and preterm premature rupture of membranes[68].

## The separate effects of IUI, subfertility, and in vitro culture

To separate the effects of ART methods and subfertility, we compared DNAm and gene expression of placentas derived from IUI ($n = 12$, $n = 10$, respectively) and SF ($n = 11$, $n = 10$, respectively) couples to controls ($n = 77$, $n = 39$, respectively) (Supplementary Data 11 and 12). When IUI placentas were compared to controls, 27 DMPs (linking to 25 genes), 13 DMRs (linking to 14 genes), and one upregulated DEG, a non-coding RNA gene *TSIX,* were observed. In SF placentas, there were no changes in DNAm, whereas three DEGs (*ATG9B, SRRM2,* and *INSR*) were downregulated compared to controls. When both IUI and SF placentas were compared to controls, 21 DMPs (19 genes), nine DMRs (eight genes), and five downregulated DEGs (*DLK1, VIM, IGFBP5, TBX2,* and *TRPC6*) were detected. Notably, decreased counts of *DLK1* were also observed in SF placentas (Fig. 1d), suggesting that downregulation of *DLK1* associates with subfertility.

To separate the effects of in vitro culture from the effects of hormonal treatments and subfertility, we compared IUI, SF, and control placentas to the ART placentas. According to this comparison, only 37 DMPs linking to 26 genes and 47 DMRs linking to 53 genes were associated with in vitro culture in DNAm analysis (Fig. 1f, Supplementary Data 11). Interestingly, among these genes were *APC2, KIFC2, MUC5B, CELF5, KNDC1,* and *FAM83H-AS1,* which have previously been associated with IVF in the first-trimester placenta[62]. mRNA-seq analysis revealed 13 DEGs associating with in vitro culture (Fig. 1g, Supplementary Data 12).

## ART-associated changes in IVF and ICSI placentas

When IVF ($n = 50$) and ICSI ($n = 30$) placentas were compared separately to controls ($n = 77$), 230 IVF-associated DMPs linking to 148 genes (Fig. 2a, Supplementary Data 13) and 395 DMRs linking to 424 genes were detected (Supplementary Data 14) (general characteristics in Supplementary Data 1). The IVF-associated DMRs and the pathways in which they clustered were similar to ART result with additional terms such as insulin secretion (GO:BP), maturity onset diabetes of the young, and steroid biosynthesis (KEGG) ($P < 0.05$) (Fig. 2c, Supplementary Data 15). The ICSI placentas associated with 12 DMPs linking to 9 genes (Fig. 2b, Supplementary Data 16). Only three DMPs were common between the IVF and ICSI placentas and when the IVF placentas were compared to ICSI, no DMPs or DMRs were detected (Fig. 2f, Supplementary Data 17).

mRNA-seq analysis for the IVF ($n = 38$) and control ($n = 39$) placentas showed 14 DEGs (10 down- and 4 upregulated), whereas a higher number,

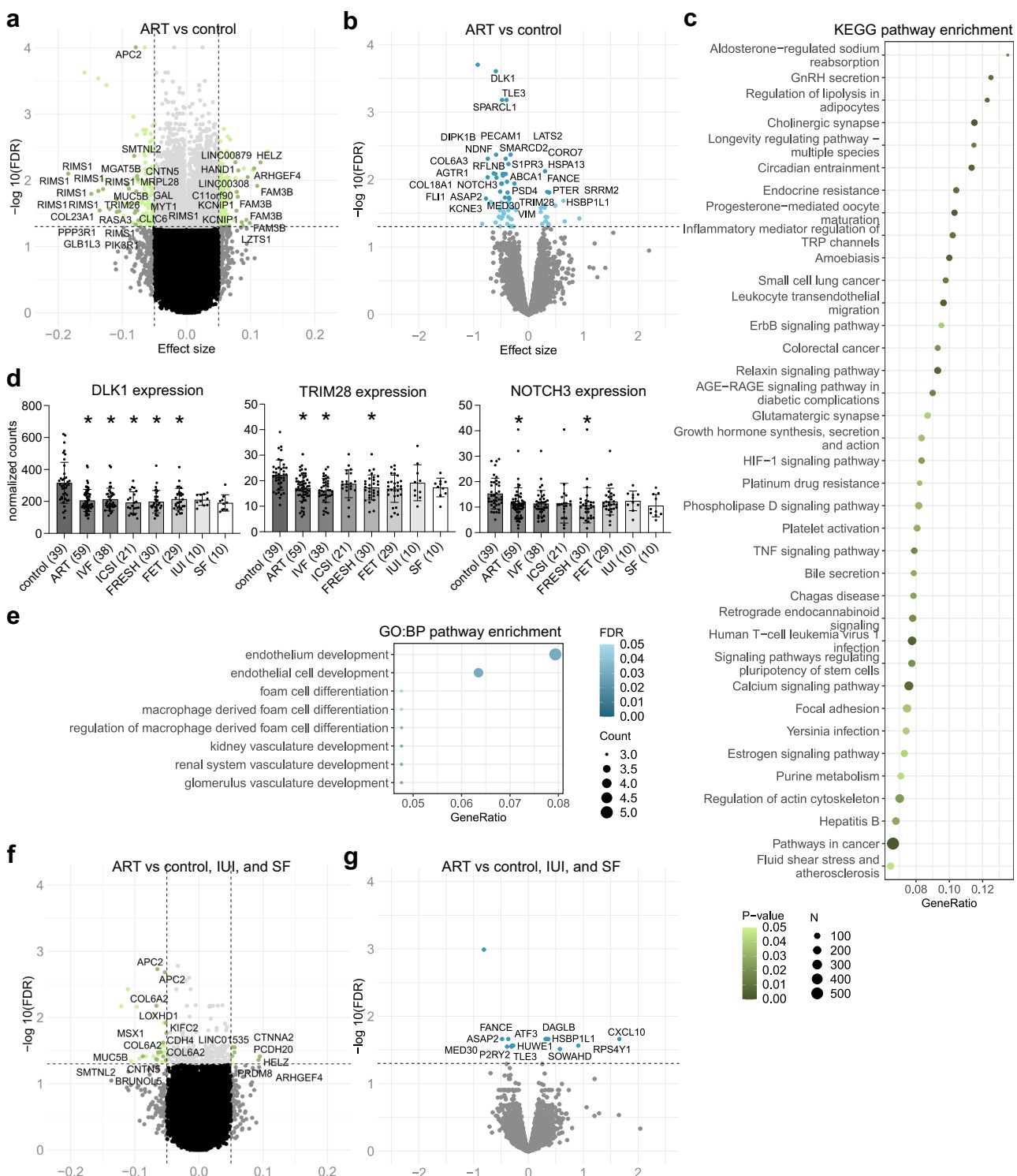

**Fig. 1 | ART-associated differential DNAm and ART-, IUI-, SF-, and in vitro culture-associated differential gene expression. a** Volcano plot showing the distribution of associations between CpG sites and ART. All DMPs (FDR < 0.05, Δβ ≤ −0.05 and Δβ ≥ 0.05) are shown in green, and for visualization, the DMPs with the largest effect sizes (Δβ ≤ −0.075 and Δβ ≥ 0.075) are labeled based on UCSC RefGene Name. **b** Volcano plot showing the distribution of associations between mRNA expression and ART. The 28 most significant DEGs are labeled for visualization including only confirmed genes. **c** Significantly enriched terms identified in KEGG enrichment analysis for ART-associated DMRs (P <0.05). **d** Normalized mRNA-seq counts of *TRIM28, DLK1*, and *NOTCH3* genes in control, ART, IVF, ICSI, FRESH, FET, IUI, and SF placental samples. Data presented as mean ± SD.

*FDR < 0.05 in genome-wide mRNA-seq analysis compared to controls. **e** Significantly enriched terms identified in the GO:BP enrichment analysis for ART-associated DEGs (FDR-corrected *q*-value < 0.05). **f** Volcano plot showing the distribution of associations between CpG sites and in vitro culture. All DMPs (FDR < 0.05, Δβ ≤ −0.05 and Δβ ≥ 0.05) are shown in green, and labeled based on UCSC RefGene Name. **g** Volcano plot showing the distribution of associations between mRNA expression and in vitro culture. All significant DEGs are labeled including only confirmed genes. In volcano plots, horizontal line marks FDR 0.05 and vertical line marks effect size ±0.05. Control *n* = 77, ART *n* = 80, IUI *n* = 12, and SF *n* = 11 in DNAm a*n*d control *n* = 39, ART *n* = 59, IVF *n* = 38, ICSI *n* = 21, IUI *n* = 10, and SF *n* = 10 in mRNA-seq analyses.

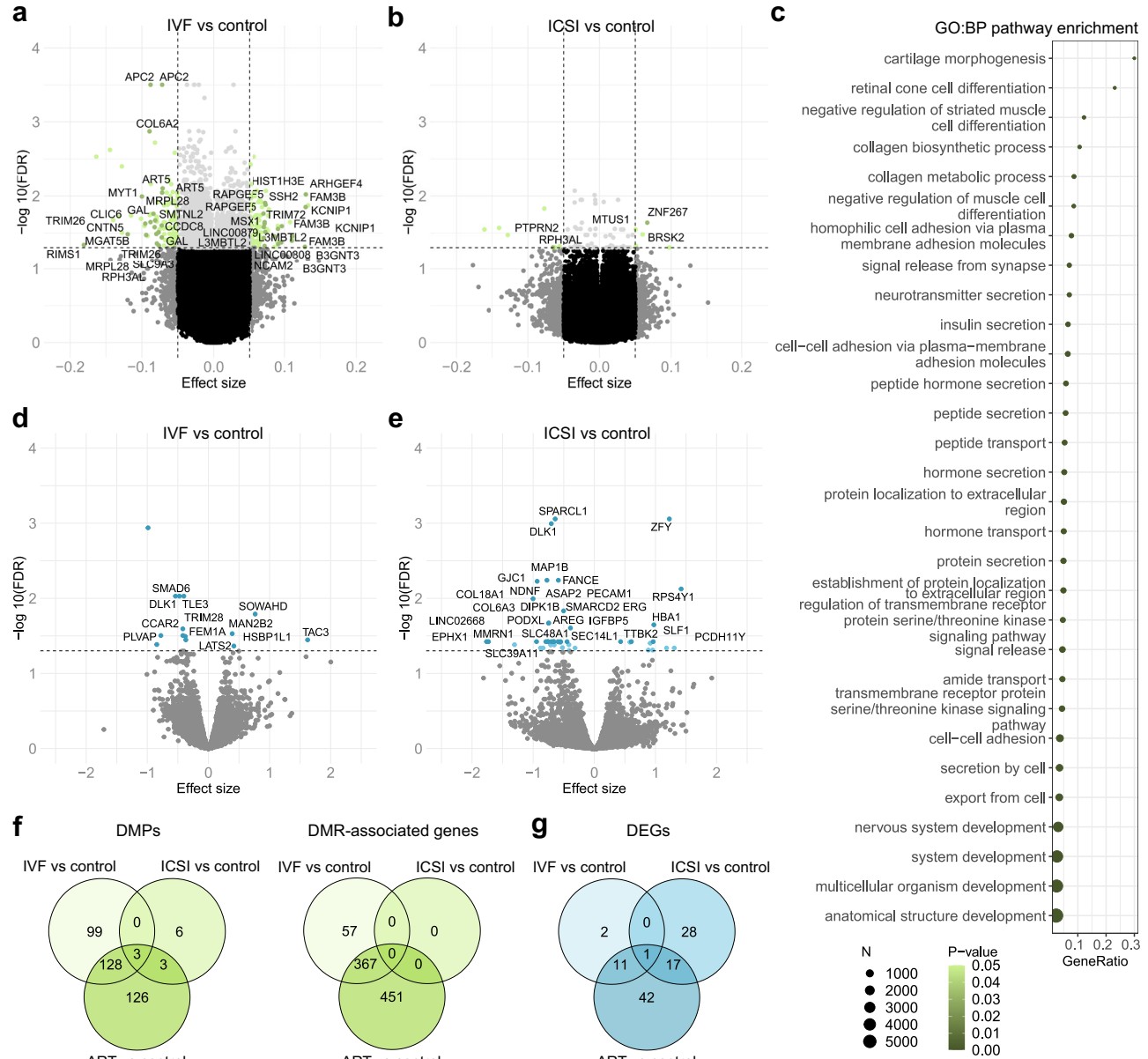

**Fig. 2 | IVF- and ICSI-associated differential DNAm and gene expression.**
**a** Volcano plot showing the distribution of associations between CpG sites and IVF.
All DMPs (FDR < 0.05, $\Delta\beta \leq -0.05$ and $\Delta\beta \geq 0.05$) are shown in green, and for
visualization, the DMPs with the largest effect sizes ($\Delta\beta \leq -0.07$ and $\Delta\beta \geq 0.07$) are
labeled based on UCSC RefGene Name. **b** Volcano plot showing the distribution of
associations between CpG sites and ICSI. DMPs are labeled based on UCSC
RefGene Name. **c** The 30 most significantly enriched terms identified in the GO:BP
enrichment analysis for IVF-associated DMRs ($P < 0.05$). **d** Volcano plot showing
the distribution of associations between mRNA expression and IVF. All significant

DEGs are labeled including only confirmed genes. **e** Volcano plot showing the
distribution of associations between mRNA expression and ICSI. The 28 most
significant DEGs are labeled for visualization. **f** Venn diagram showing the number
of ART-associated DMPs as well as DMR-associated genes, which are in common
between IVF and ICSI placentas. **g** Venn diagram showing the number of ART-
associated DEGs, which are in common between IVF and ICSI placentas. In volcano
plots, horizontal line marks FDR 0.05 and vertical line marks effect size ±0.05.
Control $n = 77$, IVF $n = 50$, and ICSI $n = 30$ in DNAm and control $n = 39$, IVF
$n = 38$, and ICSI $n = 21$ in mRNA-seq analyses.

46 genes (34 down- and 12 upregulated), was observed in the ICSI placentas
($n = 21$) (Fig. 2d, e, Supplementary Data 17 and 18). *DLK1* was the only
common DEG between IVF and ICSI placentas. Interestingly, the upre-
gulated genes *ZFY, RPS4Y1, PCDH11Y*, and *DDX3Y* in the ICSI placentas
locate on chromosome Y and the expressions of them all correlated sig-
nificantly with each other in ART male placentas, but not in the controls
(Supplementary Data 9). Since male infertility was a common reason
(44.4%) for the infertility among the analyzed ICSI male samples (Supple-
mentary Data 2) the upregulated genes *DDX3Y*[69], *ZFY*[70], and *PCDH11Y*[71],
which all have been associated with male infertility in previous studies, are
notable. Furthermore, the ICSI-associated DEGs *MAP1B, HBA1, EPHX1,
HBB*, and *HBG2* enriched in response-to-toxic-substance pathway are also

interesting (FDR-corrected $q$-value < 0.05), considering that environmental
exposures have been associated with male infertility in previous studies
(Supplementary Data 19). Epoxide hydrolases, such as *EPHX1*, are involved
in detoxifying and excreting the environmental chemicals which are asso-
ciated with decreased semen quality and male infertility[72].

## ART-associated changes in FRESH and FET placentas
To study the effects of embryo freezing on DNAm, we separately compared
FRESH ($n = 42$) and FET ($n = 38$) placentas to the controls ($n = 77$). A total
of 119 FRESH-associated DMPs were annotated to 110 genes and 67 FET
DMPs to 54 genes (Fig. 3a–c, Supplementary Data 20–22). Furthermore,
144 FRESH-associated DMRs linking to 146 genes and 46 FET DMRs

linking to 47 genes were observed (FDR < 0.05) (Fig. 3c, Supplementary Data 22−24). GO enrichment analysis of FRESH DMRs revealed BPs involved in pathways such as cell-cell adhesion (FDR < 0.05), nervous system development, hormone secretion and transport as well as the regulation of insulin secretion involved in cellular response to glucose stimulus ($P < 0.05$). FET DMRs clustered to the regulation of GTPase activity, cytosol to endoplasmic reticulum transport, and regulation of astrocyte differentiation ($P < 0.05$) (Supplementary Data 25 and 26, respectively).

mRNA-seq analysis for the FRESH ($n = 30$), FET ($n = 29$), and control ($n = 39$) placentas showed 41 FRESH-associated DEGs (35 down- and six upregulated) and only six FET DEGs (five down- and one upregulated) (Fig. 3d–f, Supplementary Data 22 and 27). Downregulated *DLK1* came up from both FRESH and FET placentas, but *NOTCH3* and *TRIM28* were significantly downregulated only in FRESH placentas (Fig. 1d). The only upregulated FET-associated DEG was *dermatan sulfate epimerase* (*DSE*), which overexpression has been observed in several cancers and which associates with active angiogenesis, invasion, and proliferation[73].

Finally, to exclude potential ICSI procedure-associated variation, we compared FRESH ($n = 25$) and FET ($n = 25$) placentas derived only from the IVF procedure to the controls ($n = 77$). A total of 21 DMPs and 25 DMRs were associating with IVF-FRESH placentas, whereas 35 DMPs and 15 DMRs associated with IVF-FET (Fig. 3g, h, Supplementary Data 28 and 29, respectively). Genes associating with DMPs and DMRs specifically in IVF-FET placentas were *RAPGEF5, ARHGEF4, GAL,* and *KCP*. Only one DEG in IVF-FRESH placentas ($n = 19$), downregulated *CBX6* (log fold change = −0.97, FDR = 0.01), and no DEGs in the IVF-FET placentas ($n = 19$) were observed when compared to controls ($n = 39$).

## Sex-specific differences in ART-derived placentas

To identify sex-specific effects of ART, we performed DNAm analyses including sex chromosomes for the placentas of male and female newborns separately. The GWAM overall level was slightly higher in male placentas compared to females, which is consistent with a previous study[31] (Supplementary Figs. 7, 8). A higher GWAM overall level was observed in ICSI male placentas ($n = 12$) compared to control males ($n = 42$), and significantly higher compared to IVF males ($n = 24$) ($d = 0.892, P = 0.035$). Furthermore, ICSI male placentas had significantly higher GWAM level at genomic locations 5'UTR ($d = 0.972, P = 0.018$) and S_Shore ($d = 1.05, P = 0.021$) compared to control males as well as at 5'UTR ($d = 0.964, P = 0.037$), S_Shelf ($d = 0.942, P = 0.009$), and S_Shore ($d = 1.088, P = 0.018$) compared to IVF males. By contrast to the males, differences in GWAM overall level were not observed between control ($n = 35$), IVF ($n = 26$), and ICSI ($n = 18$) female placentas. However, the IVF female placentas had significantly higher GWAM at the gene body compared to controls ($d = 0.607, P = 0.047$) as well as higher GWAM at S_Shelf compared to ICSI females ($d = 0.917, P = 0.011$). Moreover, a trend of higher GWAM in both male and female FET placentas compared to FRESH and controls was observed (Supplementary Fig. 7 and 8). FET female placentas ($n = 16$) had significantly lower GWAM at 1stExon compared to FRESH females ($n = 28$) ($d = −0.929, P = 0.032$). GWAM levels were compared by using One-way ANOVA followed by Tukey's HSD.

In ART male placentas, there were 12 DMPs and 17 DMRs compared to control males ($n = 36$, control: $n = 42$), whereas 18 DMPs and five DMRs were observed in ART females (ART: $n = 44$, control: $n = 35$) (Supplementary Data 30). Interestingly, mRNA-seq for male placentas (ART: $n = 29$, control: $n = 19$) revealed 21 DEGs (18 down, 3 up) (FDR < 0.05) clustering in pathways of endothelial and epithelial cell development in GO:BP enrichment analysis, and none DEGs in the ART females (ART: $n = 30$, control: $n = 20$) (Supplementary Data 31). In IVF male placentas ($n = 20$), five DMPs and five DEGs (downregulated *PECAM1, SPARCL1, DLK1,* and *S1PR3* and upregulated *LYPLA1)* were observed compared to control males ($n = 19$), and in IVF female placentas ($n = 26$) 24 DMPs and 12 DMRs were compared to control females ($n = 35$). Despite of the small sample size in sex-specific analysis of ICSI samples, 16 DEGs were observed in ICSI male placentas ($n = 9$, control males: $n = 19$), including five

upregulated genes, *DDX3Y, EIF1AY, ZFY, RPS4Y1,* and *PCDH11Y*, locating all on chromosome Y. In female placentas ($n = 12$, control females $n = 20$), six DEGs were observed including downregulated *CASP2* and upregulated *TFPI2, CSH1, CGA, PSG2,* and *PSG3*.

When FRESH and FET placentas were analyzed sex-specifically, only one downregulated DEG (*SPARCL*), was observed in FRESH male placentas ($n = 11$, control males: $n = 19$) and seven DMPs in the FRESH female placentas ($n = 28$, control females: $n = 35$) (Supplementary Data 30 and 31). In FET male placentas, there were 11 DEGs clustering in GO:BP pathway term protein refolding (*HSPA13, HSPA5*) ($n = 18$, control males: $n = 19$), while four DMPs were observed in FET female placentas (Supplementary Data 30 and 31).

## Associations between placental gene expression and phenotypes

Potential correlations between *TRIM28, NOTCH3,* and *DLK1,* and phenotypes of newborns and their mothers were calculated by Spearman rank correlation (Supplementary Data 32). When all samples were examined, there was a weak correlation between placental *NOTCH3* expression and BW (SD) ($r = 0.272, P = 0.003, n = 118$), and it was driven by male samples ($n = 58$) in which the correlations between *NOTCH3* and BW (SD) ($r = 0.414, P = 0.001$) as well as *NOTCH3* and BL (SD) ($r = 0.369, P = 0.004$) were observed. Furthermore, *NOTCH3* expression correlated with the BW (SD) ($r = 0.41, P = 0.01$) and BL (SD) ($r = 0.386, P = 0.015$) of controls ($n = 39$), with BW (SD) ($r = 0.441, P = 0.046$), BL (SD) ($r = 0.503, P = 0.02$) and placental weight ($r = 0.463, P = 0.035$) of ICSI ($n = 21$), as well as with BW (SD) ($r = 0.419, P = 0.024$) and BL (SD) ($r = 0.441, P = 0.017$) of FET newborns ($n = 29$). The correlation between *NOTCH3* and BL (SD) among the ICSI samples was driven by male newborns.

In all male samples, there was a negative correlation between *TRIM28* expression and maternal age ($r = −0.339, P = 0.009, n = 58$) and in all the female samples, *TRIM28* correlated weakly with GA ($r = 0.257, P = 0.048, n = 60$). Furthermore, *TRIM28* expression correlated negatively with maternal weight gain during pregnancy in the control ($r = −0.418, P = 0.01, n = 37$) and ART ($r = −0.295, P = 0.03, n = 54$) placentas. In the ART placentas it was driven by ICSI ($r = −0.596, P = 0.007, n = 19$), specifically by ICSI female placentas ($r = −0.655, P = 0.029, n = 11$). Also, placental *TRIM28* expression correlated negatively with maternal weight gain ($r = −0.388, P < 0.05, n = 26$) and positively with maternal BMI in the FRESH placentas ($r = 0.363, P = 0.048, n = 30$).

Placental *DLK1* expression correlated with ART female BL (SD) ($r = 0.394, P = 0.031, n = 30$) driven by ICSI females ($r = 0.676, P = 0.016, n = 12$), and negatively with maternal weight gain during the pregnancy in IVF females ($r = −0.618, P = 0.014, n = 15$).

## ART- and subfertility-associated placental imprinting

Finally, we focused on genomic imprinting since imprinting disorders have been associated with ART in several previous studies[74]. A total of 18 ART- and/or ART subgroup-associated DMRs on the microarrays are annotated to the imprinted or predicted imprinted genes[50] (ART: $n = 80$, IVF: $n = 50$, ICSI: $n = 30$, FRESH: $n = 42$, FET: $n = 38$, control: $n = 77$) (Supplementary Data 33). Seven of these (*PRDM16, FBRSL1, HOXA3, HOXC4, OBSCN*[12], *TP73,* and *HOXA5*[21]) have been associated with ART also in previous EWASs. To focus specifically on ICRs, which are established in the germline to control imprinted gene expression, we compared 826 CpGs on ICRs compiled by Ochoa and colleagues[54] between ART and control placentas and observed significant differences in 148 sites (108 hypo- and 40 hypermethylated) ($P < 0.05$, Student's $t$-test) (Supplementary Data 33). After multiple testing correction for these 826 sites, 25 CpG sites remained significant (Bonferroni adjusted $P < 0.05$).

Interestingly, in the current study, we observed IVF-associated DMR linked to *DIO3* (Supplementary Data 33) as well as ART-associated downregulation of paternally expressed *DLK1*, which both locate in a paternally imprinted *DLK1-DIO3* locus. Since also the downregulation of *TRIM28*, the stabilizer of ICR DNAm at the *DLK1-DIO3* locus[75], was

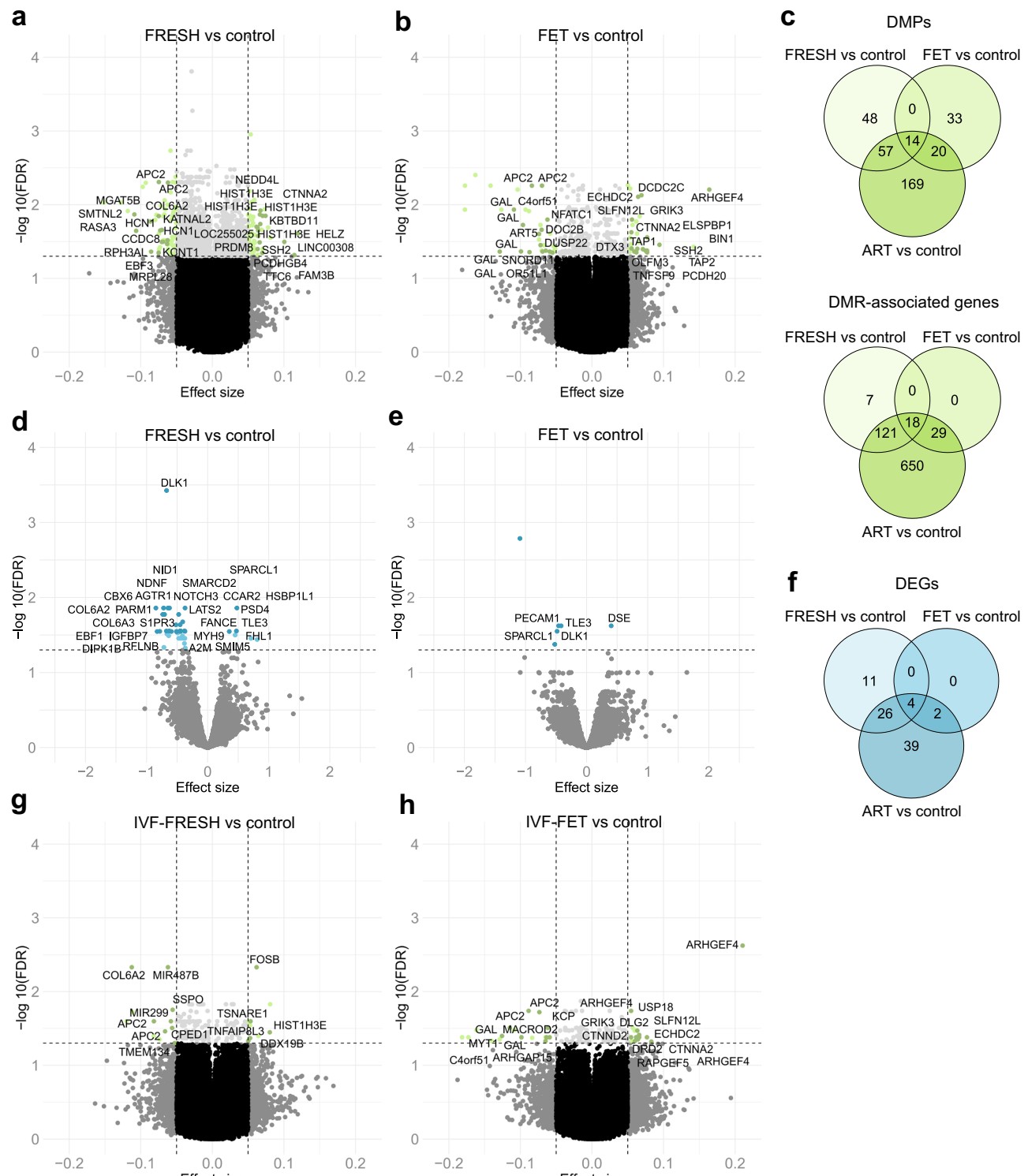

**Fig. 3 | FRESH- and FET-associated differential DNAm and gene expression.**
**a** Volcano plot showing the distribution of associations between CpG sites and FRESH. DMPs (FDR < 0.05, $\Delta\beta \leq -0.05$ and $\Delta\beta \geq 0.05$) are shown in green, and for visualization, the DMPs with the largest effect sizes ($\Delta\beta \leq -0.06$ and $\Delta\beta \geq 0.06$) are labeled based on UCSC RefGene Name. **b** Volcano plot showing the distribution of associations between CpG sites and FET. DMPs with the largest effect sizes ($\Delta\beta \leq -0.055$ and $\Delta\beta \geq 0.055$) are labeled based on UCSC RefGene Name for visualization. **c** Venn diagram showing the number of ART-associated DMPs as well as DMR-associated genes, which are in common between FRESH and FET placentas. **d** Volcano plot showing the distribution of associations between mRNA expression and FRESH. The 26 most significant DEGs are labeled for visualization including only confirmed genes. **e** Volcano plot showing the distribution of associations between mRNA expression and FET. All significant DEGs are labeled including only confirmed genes. **f** Venn diagram showing the number of ART-associated DEGs, which are in common between FRESH and FET placentas. **g** Volcano plot showing the distribution of associations between CpG sites and IVF-FRESH. All DMPs are labeled. **h** Volcano plot showing the distribution of associations between CpG sites and IVF-FET. All DMPs are labeled. Control $n = 77$, ART $n = 80$, IVF $n = 50$, ICSI $n = 30$, FRESH $n = 42$, FET $n = 38$, IVF-FRESH $n = 25$, and IVF-FET $n = 25$ in DNAm and control $n = 39$, FRESH $n = 30$, and FET $n = 29$ in mRNA-seq analyses.

observed in ART placentas, we examined DNAm at this ICR of ART (IVF: $n$ = four, ICSI: $n$ = four), SF ($n$ = four), and control ($n$ = six) placentas by traditional bisulfite sequencing (Fig. 4a). The analyzed ICR sequence is next to the repeated sequence motifs, which contribute to the DNAm of the ICR[75]. Maternal and paternal alleles were distinguished by two SNPs (rs1884539 and rs75998174) when possible. Interestingly, although we did not see highly methylated paternal and demethylated maternal allele in this specific ICR sequence, we observed significantly decreased total DNAm in the ICR of SF placentas compared to control (CpG1: $d = -2.675$, $P = 0.003$, CpG2: $d = -2.305$, $P = 0.007$, One-Way ANOVA followed by Bonferroni *post hoc*) and ART placentas (CpG1: $d = -2.507$, $P = 0.002$, CpG7: $d = -1.857$, $P = 0.047$, One-Way ANOVA followed by Bonferroni *post hoc*) (Fig. 4b, Supplementary Fig. 2, Supplementary Data 34). However, we did not observe significantly decreased DNAm in the ART placentas compared to controls.

## Discussion

In this study, we performed the first genome-wide DNAm and gene expression analyses of human placentas from ART pregnancies as far as we are aware. Those pregnancies are not a homogeneous group, but differ from each other due to the ART procedures, causes of underlying infertility, and sex of the child. By combining the omics data to the phenotypic information and conducting method- and sex-specific analyses, it was possible to perceive the associations between molecular alterations and phenotypic features. There were only a low number of in vitro culture-associated changes or common ART procedure-associated alterations in the IVF and ICSI placentas, suggesting a strong impact of the underlying subfertility on the DNAm and gene expression rather than the effects of the in vitro culture. Interestingly, some of the ART-associated DNAm changes in the placenta observed in the current study, have been reported previously in the EWASs of blood samples. These findings confirmed our analyses and may reflect early epigenetic effects of ART procedures, such as hormonal treatment, that remain in the cells' epigenetic mitotic memory. However, the effects of subfertility or fertility-associated genetic variation on DNAm cannot be excluded in this study. Furthermore, we confirmed previous findings about the procedure-specific effect of FET and observed heavier FET placentas compared to the FRESH placentas. Our observation about the lower amount of DNAm and gene expression changes in the FET placentas compared to FRESH was consistent with this. Although freezing might be a challenging environmental insult for the embryo, it enables to avoid the

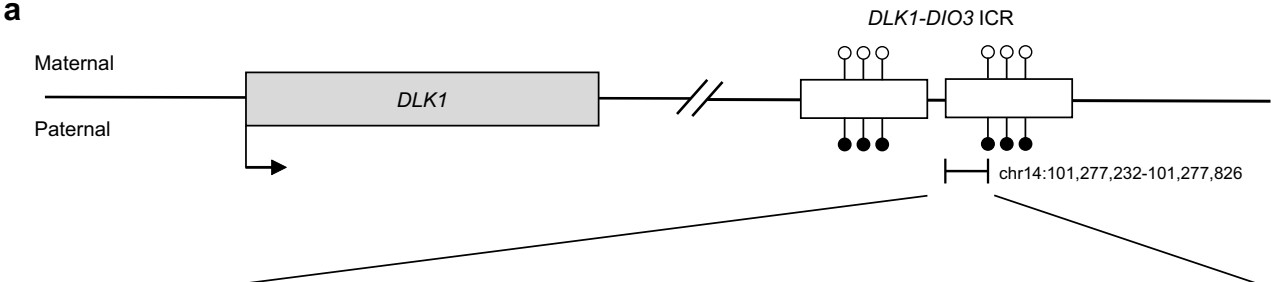

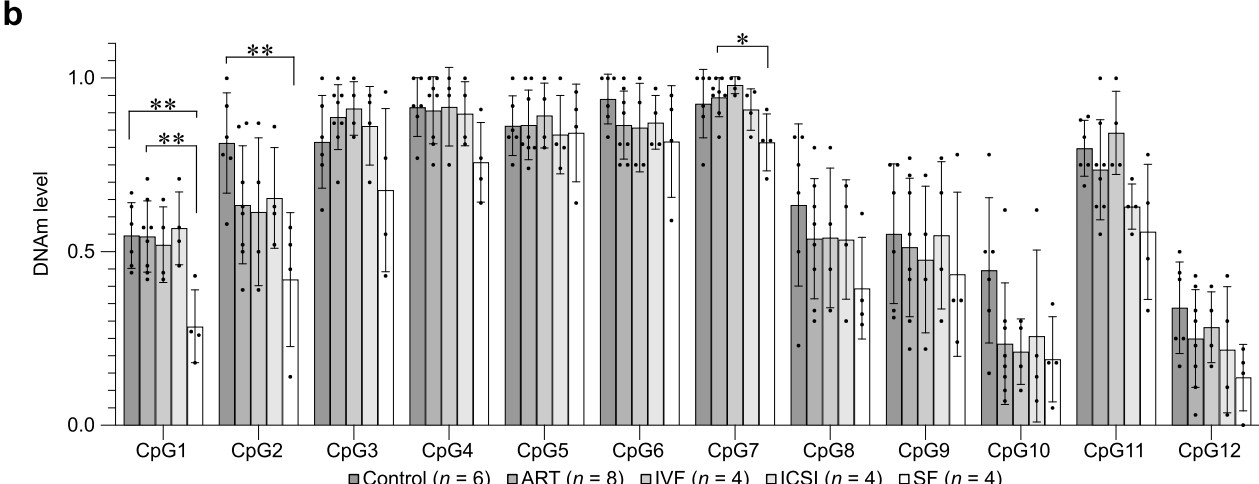

**Fig. 4 | ART- and SF-associated DNAm profiles at *DLK1-DIO3* ICR. a** Schematic figure about *DLK1* and *DLK1-DIO3* ICR based on Kagami et al.[99]. The chromosomal region chr14:101,277,232-101,277,826 at the ICR, containing repeated sequence with nine underlined motifs (chr14:101,277,232-101,277,411) and PCR amplified sequence (chr14:101,277,375-101,277,826), is presented. Primers spanning the amplicon are highlighted in gray and the covered 12 CpG sites and the SNPs (rs1884539A/G and rs75998174A/G) used to distinguish paternal and maternal alleles are bolded. **b** DNAm levels at 12 CpG sites in *DLK1-DIO3* ICR in control ($n$ = 6), ART ($n$ = 8), IVF ($n$ = 4), ICSI ($n$ = 4), and SF ($n$ = 4) placentas. Differences in DNAm at the CpG sites were calculated between control, ART, and SF placentas. Data presented as mean ± SD. *$P < 0.05$ and **$P < 0.01$, One-Way ANOVA followed by Bonferroni *post hoc*.

hormonal disturbances caused by ovarian stimulation in FRESH. Nearly all (82%) of the FET pregnancies in our study have been started by natural cycle, which could explain the differences between FRESH and FET placentas.

In some studies, the more invasive ICSI procedure has been associated with a higher risk of birth defects[76], which is concerning due to the increased use of this method. In the current study, more changes in gene expression were observed in the ICSI placentas compared to IVF. Male factor infertility was a prominent reason for ICSI in this study, which could explain the observed alterations. This can be seen in a small number of common ART-associated genes in IVF and ICSI placentas as well as in many male infertility-associated genes on the Y chromosome in the ICSI male placentas. Indeed, in addition to information about the placenta itself, placenta could offer a molecular window to early development[77–79], even to the germ cells. The origin of these early alterations and their potential to provide future biomarkers of infertility should be studied. In addition to the infertility-associated genes in Y chromosome, several sex-specific ART-associated alterations were observed in the current study. The number of sex-specific studies so far is surprisingly small and more studies with larger sample sizes are needed to verify these discoveries.

In addition to confirming previous ART-associated alterations observed in blood and placental samples, we found new candidate genes *DLK1*, *TRIM28*, and *NOTCH3* for ART-associated phenotypes. By performing cell type fragment analysis, we observed a significantly lower number of stromal cells in the ART placentas compared to controls. Interestingly, it has been suggested that differences in placental mesenchymal stromal cells could lead to impaired vascular development and consequently to restricted growth[80,81]. This is supported by our gene expression analysis, in which the most prominent ART-associated changes were linked to vasculogenesis, and where the downregulation of *NOTCH3* and *TRIM28*, both involved in the regulation of angiogenesis through the VEGFR-DLL4-NOTCH signaling circuit[82], was observed. Furthermore, *NOTCH3* expression correlated with BW (SD) of control newborns, with BW (SD), BL (SD), and placental weight of ICSI male newborns, as well as BW (SD) and BL (SD) of FET newborns. Since *NOTCH3* and *TRIM28* are not significantly downregulated in placentas of FET newborns—whose size is near to the control newborns—the downregulation of *NOTCH3* could explain the higher risk of LWB and SGA associated with FRESH. Interestingly, among the control and FET groups the maternal hormonal function is not affected: in most cases the embryo was transferred in a natural cycle without the hormonal disturbances caused by ovarian stimulation. The role of hormones brought forth also in the enrichment analysis, where ART-associated DMRs linked to terms such as growth hormone synthesis, secretion, and action, as well as GnRH secretion, and estrogen signaling pathway. Recently, it has been shown that *TRIM28* modulates estrogen receptor α and progesterone receptor signaling, which regulate endometrial cell proliferation, decidualization, implantation, and fetal development[64]. Considering all these observations, the functional links between *NOTCH3*, *TRIM28*, and the ART-associated cardiovascular phenotype should be studied. Also, the separate effects of hormonal ovarian stimulation in ART protocols and subfertility-associated impaired hormonal function on early development need to be clarified in the future.

Along with *TRIM28* and *NOTCH3*, one of our most interesting discoveries in the ART placentas concerned the prominent downregulation of imprinted *DLK1* as well as the pathways of insulin secretion and maturity onset diabetes of the young, which came forth in the DNAm analyses. The developmentally essential *DLK1* is a part of evolutionary conserved Delta-Notch pathway, and it inhibits adipogenesis by controlling the cell fate of adipocyte progenitors[83,84]. According to animal studies, mice without paternally expressed *DLK1* exhibit growth retardation and obesity while *DLK1* overexpression generate decreased fat mass, diet-induced obesity resistance, and reduced insulin signalling[85,86]. In human, mutations in *DLK1* gene have been reported as a cause of central precocious puberty associated with obesity and metabolic syndrome with undetectable DLK1 serum levels[87]. Furthermore, in Temple syndrome, where *DLK1* expression is

downregulated due to the maternal uniparental disomy of the imprinted *DLK1* locus, phenotypic features such as prenatal growth failure, short postnatal stature, female early onset puberty, and truncal obesity have been observed[88]. Since increased risks for SGA, rapid postnatal growth, and female early onset puberty[6] have all been associated with ART children, *DLK1* is a plausible candidate gene for the phenotypic effects associated with ART.

In addition to the downregulation of *DLK1* expression, we observed ART-associated downregulation of *TRIM28*, which is needed for the stability of DNAm in the imprinted regions[89]. We observed decreased DNAm in several ICRs – common binding sites of *TRIM28*—suggesting that the machinery, which normally stabilizes the DNAm of ICRs globally could be impaired. This could explain the high variability in ART-associated imprinted genes observed in previous studies. More specific examination of *DLK1-DIO3* locus showed decreased DNAm of the ICR in SF placentas. This suggests that the instability of DNAm at this locus associates with subfertility, which is consistent with the decreased *DLK1* expression in the ART, SF, and IUI placentas observed in the current study. Consistent with this, associations between subfertility and decreased placental *DLK1* expression[19], as well as between mutated *DLK1* and infertility[63], has been reported in previous studies. Unexpectedly, decreased DNAm was not observed in the ART placentas. This can be explained by the conditions of in vitro culture, which has been found to cause aberrant hypermethylation in *DLK1-DIO3* locus[90]. Moreover, *TRIM28* can regulate the expression of imprinted *DLK1* in several ways[89,91] and therefore the interaction of these two genes should be examined by functional studies. Also, the effect of ageing on *TRIM28* expression and its associations with both *DLK1* and *NOTCH3* should be studied profoundly, due to the previous association between *TRIM28* and testicular degeneration and age-dependent infertility in mouse[92] as well as our observed negative correlation between placental *TRIM28* expression and maternal age.

These findings are also interesting in the perspective of fetal programming. According to the programming hypothesis, adverse conditions such as prenatal malnutrition in the critical developmental period may permanently program physiological processes. If there is plenty of food available postnatally, this programming may prove inappropriate and cause metabolic alterations leading to adult diseases such as obesity, T2D, metabolic syndrome, and cardiovascular disease[93,94]. Both, *TRIM28* and *DLK1* are plausible candidate genes for fetal programming, owing to their association with obesity[95,96] as well as previously reported *TRIM28*'s role in the regulation of *DLK1* in adipogenesis[91]. Previously, Cleaton and colleagues[97] showed in mouse that both *Dlk1* expression in maternal tissues and circulating DLK1 derived from the fetus are necessary for appropriate maternal metabolic adaptations to pregnancy by allowing the switch to fatty acid use when resources are limited during fasting. Also, they reported that the circulating DLK1 level predicts embryonic mass in mouse and lower DLK1 levels were strongly associated with high-resistance patterns of flow in the umbilical artery and slow abdominal circumference growth velocity in a human cohort. Furthermore, DLK1 level enables to differentiate healthy SGA from pathologically small infants. Our observation about the downregulated placental *DLK1* and *TRIM28* as well as their negative correlation with maternal weight gain during the pregnancy suggest that they did not effectively inhibit maternal adipogenesis, resulting in increased maternal weight and impaired maternal metabolic adaptations to the pregnancy. This negative correlation was seen particularly in our IVF and ICSI female as well as FRESH placentas, in the subgroups where the relatively smaller placentas and newborns were derived from. The switch to fatty acid use in maternal tissues during fasting could be impaired, potentially causing restricted fetal growth and a higher risk for adult disorders in the future. However, in the ART-associated phenotype the increased risk for LBW, SGA, rapid post-natal growth, and metabolic disorders would be a consequence of downregulation of *DLK1* caused by subfertility associated disorder, not a consequence of poor maternal nutrition as in the example of fetal programming. Notably, significant associations between subfertility and LBW have been observed in previous studies[98].

We are aware of the limitations in this study. Owing to the limited sample size, we were not able to perform sex-specific analyses for all the subgroups, or focus on the details of the procedures, such as the length of embryo culture, hormonal treatments, or freezing methods. Furthermore, due to the methodological limitations, we were not able to perform cell-type adjustment for mRNA-seq data.

In summary, the observed molecular changes in genome-wide DNAm and gene expression in the ART placentas were relatively subtle, which is a finding in line with the fact that the majority of the ART conceived children are as healthy as newborns from natural conceptions. Decreased expression of *TRIM28, NOTCH3*, and *DLK1* bring forth potential mechanisms for the metabolic and phenotypic features including growth disturbance and imprinting disorders, which have been associated with ART. Considering that the observed changes associated also with subfertility, they offer a precious insight to the molecular background of infertility.

## Data availability
The datasets supporting the conclusions of the current study (source data) are included within the article and its additional files. According to the ethical approvals and participant consents, access to the raw data is not permitted outside of approved research projects. All other data are available from the corresponding author on reasonable request.

## Abbreviations

| | |
|---|---|
| ART | Assisted reproductive technology |
| BL | Birth length |
| BMI | Body mass index |
| BP | Biological process |
| BW | Birth weight |
| CpG | Cytosine-phosphate-guanine site |
| CS | Cesarean section |
| DEG | Differentially expressed gene |
| DIO3 | Iodothyronine deiodinase 3 |
| DLK1 | Delta like non-canonical Notch ligand 1 |
| DMP | Differentially methylated position |
| DMR | Differentially methylated region |
| DNAm | DNA methylation |
| DSE | Dermatan sulfate epimerase |
| ERV | Endogenous retrovirus |
| EWAS | Epigenome-wide association study |
| FET | Frozen embryo transfer |
| FRESH | Fresh embryo transfer |
| FuGU | Biomedicum Functional Genomics Unit |
| GA | Gestational age |
| GnRH | Gonadotropin-releasing hormone |
| GO | Gene ontology |
| GWAM | Genome-wide average DNAm level |
| HC | Head circumference |
| HSD | Honestly significant difference |
| ICR | Imprinting control region |
| ICSI | Intracytoplasmic sperm injection |
| IUI | Intrauterine insemination |
| IVF | In vitro fertilization |
| LBW | Low birth weight |
| LGA | Large for gestational age |
| LINE1 | Long interspersed nuclear element 1 |
| mRNA-seq | 3'mRNA sequencing |
| N_shelf | north shelf |
| N_shore | north shore |
| NOTHC3 | Notch receptor 3 |
| PTB | Preterm birth |
| RE | Repetitive element |
| S_shore | south shore |
| S_shelf | south shelf |
| SF | Subfertile |
| SGA | Small for gestational age |
| SNP | Single nucleotide polymorphism |
| SVD | Singular value decomposition |
| T2D | Type II diabetes mellitus |
| TNF | tumor necrosis factor |
| TRIM28 | Tripartite motif-containing protein 28 |
| TSS | Transcription starting site |
| TSS1500 | 1500 bp upstream of transcription TSS |
| TSS200 | 200 bp upstream of TSS |
| UCSC | University of California, Santa Cruz |
| UTR | Untranslated region |

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

## Acknowledgements

This project was supported by the Academy of Finland (332212) and University of Helsinki (Early Career Investigator Funding, Faculty of Medicine) (N.K-A.), Finnish Cultural Foundation (00190186, 00200185,

00212573, and 00222347) (P.A.), and Research funds from Helsinki University Hospital (A.T., T.T.). Open access was funded by Helsinki University Library. We gratefully thank all families that participated in this study. We would also like to acknowledge Teija Karkkulainen, Riikka Vass, and Ira Larsen for their contribution to the recruitment of the participants, Seija Kaukoranta for her participation in data collection as well as Samuli Auvinen for his assistance with bisulphite sequencing visualization.

## Author contributions

P.A., J.V., V.S-A., C.H-G., T.T., A.T. and N.K-A. contributed to the study design. H.K., V.S-A., and A.T. recruited the study participants. P.A., J.V., K.R., E.W., H.M., C.H. and N.K-A. contributed to the sample collection and processing. P.A., K.R., I.L. and N.K-A. contributed to the laboratory experiments. P.A., J.V., K.R., I.L. and N.K-A. contributed to the data analysis. P.A., J.V. and N.K-A. drafted the manuscript. All authors contributed to the revision of the manuscript. All authors gave final approval of the version to be published.

## Competing interests

The authors declare no competing interests.
