## [Peer review file · Communications Medicine]

Genome-wide DNA methylation and gene expression in human placentas derived from Assisted Reproductive Technology

Corresponding Author: Dr Nina Kaminen-Ahola

Version 0:

Reviewer comments:

Reviewer #1

(Remarks to the Author)

This is an interesting study that confirms previous data and expands our understanding of how ART conception influences epigenetics. The study has a reasonable number of samples (80 ART and 77 controls), and the analysis of these individuals is robust – we see some validation of previous data in placenta and blood, and the gene ontology analysis seems to indicate disruption of specific pathways.

However, the sub-analyses, n=30 ICSI, n=10 IUI, etc are underpowered, and I'm not sure they are needed in this manuscript. I recommend removing this analysis, so that the first part of the paper, which is the best, is more prominent. Of course, this is up to the authors and editor.

1. In Table S4, not all probes are linked to a gene, however in Table S5 (DMRs), each DMR has a gene associated with it. Were only promoter DMRs reported, or was a different method used to link these regions to genes?
2. It is fascinating that some of the top DMRs identified in blood are also in the placenta – e.g. CHRNE and NECAB3. This is only mentioned in the results. Can the authors elaborate a bit in discussion about what this might mean. Are we looking at a SNP effect across tissues (e.g. infertility associated)? Or is it a hormone effect that influences methylation in the oocyte or early embryo?
3. In Table S10, for STIX, the mean expression in IUI is 18.6, but the standard deviation (SD) is 2,017? The same large difference is seen in other genes in this table.
4. In Table S11 and Table S13, where IVF v Controls and ICSI v Controls results are shown, the average for the IVF and ICSI groups should also be displayed in both tables. This will give the reader a sense of how different the effect of ICSI is on methylation.
5. In figure 4B, a bar graph with SD is used to represent the data. In the case of ICSI, this means that 4 data-points are presented by 3 data-points. It would be better and more transparent to show the data as dotplot showing the actual data.

Reviewer #2

(Remarks to the Author)

The paper provides a comprehensive analysis of genome-wide DNA methylation and gene expression in human placentas from pregnancies conceived through Assisted Reproductive Technology (ART). It explores the epigenetic and gene expression differences between ART and naturally conceived placentas, investigating the impact of various ART procedures and underlying subfertility. Key findings include ART-associated changes in placental development pathways, potential mechanisms underlying phenotypic features associated with ART, and insights into the molecular background of infertility. The study contributes valuable knowledge on how ART may influence placental function and development, highlighting the importance of considering ART procedures and subfertility effects in understanding the molecular alterations in placentas and their potential implications.

Strengths:

The study addresses a significant and under-explored area, providing valuable insights into the epigenetic and expression profiles of ART placentas. The methodology is robust, employing genome-wide analyses and including a comprehensive

range of ART procedures and control groups.

Specific Comments:

1. For the acronyms and abbreviations: there are some missing terms that should be collected in the list of acronyms and abbreviations such as CpG.
2. Introduction, line 95: the authors should briefly discuss evidence for the sentence "However, the results of EWASs are not consistent" and explain this.
3. Introduction, line 99: the following three paragraphs are quite important for the whole manuscript since they are showing the key contributions and novelties of this study. For the audience to better understand the contributions of this study, the authors should write a topic sentence at the beginning of line 99 and summarize the major contributions before diving into the details.
4. Introduction, line 111: a connection between the previous paragraph and this paragraph could help the introduction to be more cohesive.
5. Results Section 1: the wording in this paragraph seems to be not well-structured. It seems that the authors lumped all the facts and data presentation together and this makes the reviewer hard to find a logic chain in understanding this section. The reviewer suggests that the authors present the data in a more systematic and logical way: for example, by ART procedures or by different phenotypes. It is quite tough to understand this paragraph while hopping between different phenotypes together with between different ART procedures.
6. Results, line 315, line 325, line 336, line 342,: one particular issue related to the P-value/number of samples presentation is how they matched with different comparisons: which P-value/case number matches with which group. This seems to be a consistent headache for the reviewer to understand the data throughout the results presentation. It would be better understandable if the authors could clearly define which P-value/case number is associated with which group throughout all results sections. Simply saying "respectively" did not make the understanding easier for the audience.
7. Results, line 359: the authors need to justify why choosing the DNAm difference of >5%.
8. Meanwhile, the reviewer supposes that the "difference" is equivalent to the " $|\Delta\beta|$ larger than 0.05", and the "effect size larger than 0.05 or smaller than -0.05" in the Figures, right? If this is the case, the authors should clarify the usage of these three terms for the better understanding of the readers.
9. Results, line 370-371: the sentence needs to be rewritten to be clearer and grammatically correct.
10. Results, line 389: how did the authors define "the most prominently downregulated"? Was that based on the significance level of effect size?
11. Results, line 391-402: the majority of the contents are more suitable to be placed in the discussion section.
12. Results, line 411-412: this is an interesting finding and the authors should further discuss over this finding. Is this serendipity?
13. Results, line 434: the reviewer suggests that the authors rewrite this sentence and be more detailed.
14. Results, line 446-448: for better understanding, the authors should clearly separate the results from IVF and ICSI. It is quite confusing to understand the sentence that blend the results from both IVF and ICSI.
15. Results, line 491: the transition is abrupt. There should be a logical connection between this paragraph and the previous paragraph. A sentence of motivation would suffice.
16. Line 781: please check spelling. Should it be "Epigenome" or "Eepigenome"?
17. Table 1: the authors coded too much information using different colors in text and boxes, which makes the understanding of the table and table caption really hard. The table should be made for an easier way to understand.
18. Fig. 1: the relationship between effect size and " $\Delta\beta$ " should be clarified in the figure caption.
19. Fig. 2: The reviewer really likes the Venn plot. However, the reviewer is curious whether the overlaps are only calculated for DMPs, or did the authors also considered the positive/negative effect size? For example, are all the 5 DMPs overlapped in Fig. 2c with a positive effect size? Is it possible to incorporate the information of the "positive/negative" effect size when compared with the control group into the plot?
20. Fig. 1 – Fig. 3: it seems that the authors used different thresholds for the effect size. Please justify the choices.

Additional areas for improvement:

1. It would be beneficial to expand on the clinical implications of the observed changes, particularly how they might affect pregnancy outcomes or offspring health.
2. The study could further investigate/discuss the potential long-term effects of the identified epigenetic changes, exploring their persistence and impact beyond neonatal development.
3. While the study includes a diverse range of ART procedures, including more detailed information on the specific ART protocols used (e.g., hormone dosages, culture conditions) could enhance the understanding of procedure-specific effects.

Summary:

The paper makes a significant contribution to the field of reproductive medicine and epigenetics, offering a foundation for future research into the mechanisms by which ART may impact placental development and function. Further studies are encouraged to build on these findings, exploring the clinical relevance and potential interventions to mitigate adverse effects associated with ART.

Reviewer #3

(Remarks to the Author)

The manuscript by Auvinen et al. studied DNA methylation alterations and gene expression aberrations in placentas of ART pregnancies. The authors tried to additionally delineate the effect of various ART procedures and infertility on the placental methylome and transcriptome. This included a separate gender focused analysis to study epigenetic changes on sex chromosomes. Overall, the study is well designed and the findings will be of interest for researchers working on medically

assisted reproduction. Furthermore, the manuscript is well written and the results are well described. One of the main limitations of this study is the low sample number, a point which the authors discussed in their manuscript.

1. The authors corrected for several covariates including cell type, gender, and maternal age and pre-pregnancy BMI. As evident from Supplementary Table 1, gestational age differs significantly between ART vs control pregnancies as well as IVF vs ICSI. Did the authors consider adjusting for gestational age in their model to account for this difference?

2. The authors mentioned the following in their analysis focused on imprinted genes (Lines 593-596): "To focus specifically on ICRs, we compared 826 CpGs on ICRs compiled by Ochoa and colleagues between ART and control placentas and observed significant differences in 115 sites (90 hypo- and 25 hypermethylated) ($P < 0.05$) (Supplementary Table S32). After multiple testing correction, 16 CpG sites remained significant (Bonferroni adjusted $P < 0.05$)." Please indicate whether multiple testing correction was performed to correct for 826 measurements or for the entire set of CpG sites measured via EPIC arrays.

Minor:

a. In Lines 443-444, the authors mention: "Only five DMPs and one gene (TCN1) as well as 17 DMR-associated genes were common between the IVF and ICSI placentas and when the IVF placentas were compared to ICSI, none DMPs or DMRs were detected." Please correct the typo in the sentence by replacing "none" with "no".

b. In the results section Lines 586-587, the following is included: "Imprinted genes are a group of parent-of-origin monoallelically expressed genes, which are maintained by correctly methylated ICRs established in the germline." This definition of imprinted genes should be included in the Introduction or Discussion section with its reference and not in the Results section.

Version 1:

Reviewer comments:

Reviewer #1

(Remarks to the Author)

the authors have addressed all my comments satisfactorily.

Reviewer #2

(Remarks to the Author)

The authors have adequately addressed my comments and suggestions! The reviewer really appreciate the time and efforts!

Reviewer #3

(Remarks to the Author)

The authors have now addressed all my comments in the revised version of their manuscript.

Author's response to reviews

Title: Genome-wide DNA methylation, imprinting, and gene expression in human placentas derived from Assisted Reproductive Technology

Authors: Pauliina Auvinen, Jussi Vehviläinen, Karita Rämö, Ida Laukkanen, Heidi Marjonen-Lindblad, Essi Wallén, Viveca Söderström-Anttila, Hanna Kahila, Christel Hydén-Granskog, Timo Tuuri, Aila Tiitinen, Nina Kaminen-Ahola

Dear Reviewers,

We want to thank for all valuable comments, which significantly improved the manuscript. A revised version of the manuscript as well as the original version with marked corrections have been submitted, and point-by-point responses to the specific comments as well as changes in the manuscript are detailed below. We hope that the current version of the manuscript is acceptable for publication.

Unfortunately, we have noticed that although all samples passed the quality control steps as we stated in the materials and methods section, there were probes which should have been excluded according to the quality control thresholds. To ensure the quality of our data, we removed 45,227 weak probes from the analyses, and reanalyzed the DNAm EPIC microarray data. Now we have corrected all DNAm results, and although the main results of the manuscript remained the same, there are two notable alterations: 1) the observed DMPs were not associated with *ZFP57* and *NLRP5*, and 2) although we still saw a high number of imprinted control regions with significantly altered DNAm, the observed alteration were not enriched statistically significantly on the imprinting control regions compared to the other regions in the EPIC array. We have corrected and reorganized figures 1-3 according to the results.

However, in these new analyses we were able to use in addition to mRNA-seq data also DNAm data from IUI and SF subgroups, and therefore gather more profound information about the *in vitro* culture- and subfertility-associated DNAm changes (see “The separate effects of IUI, subfertility, and *in vitro* culture” section). We also observed a new IVF-associated DMR linking to *DIO3* at the *DLK1-DIO3* locus, which strengthen the role of this locus in ART and subfertility.

Furthermore, in addition to the changes we have done based on reviewers' comments and reanalyzed DNAm results, we have shortened the end of the imprinting part in the results (rows 603-604: “However, we did not observe significantly decreased DNAm in the ART placentas compared to controls.”) and added “Unexpectedly, decreased DNAm was not observed in the ART placentas. This can be explained by the conditions of *in vitro* culture, which has been found to cause aberrant hypermethylation in *DLK1-DIO3* locus⁹⁰.” to the discussion (rows: 698-701). Also, we have added more detailed information to the abstract about the “downregulated” expression of *TRIM28* and *NOTCH3*, a new reference to the legend of figure four, and we have removed ICSI phenotype part from the discussion (more details about ICSI phenotype can be found in our response to the first comment of Reviewer#1).

Furthermore, we have re-evaluated all authorships. MSc Pauliina Auvinen is the first author and MSc Jussi Vehviläinen is the second author. All the authors have accepted all these changes.

Sincerely,
Nina Kaminen-Ahola
21.5.2024, Helsinki, Finland

Reviewers' comments:

Reviewer #1 (Remarks to the Author):

This is an interesting study that confirms previous data and expands our understanding of how ART conception influences epigenetics. The study has a reasonable number of samples (80 ART and 77 controls), and the analysis of these individuals is robust – we see some validation of previous data in placenta and blood, and the gene ontology analysis seems to indicate disruption of specific pathways.

However, the sub-analyses, n=30 ICSI, n=10 IUI, etc are underpowered, and I'm not sure they are needed in this manuscript. I recommend removing this analysis, so that the first part of the paper, which is the best, is more prominent. Of course, this is up to the authors and editor.

Thank you for this comment. We agree that the number of samples in some subgroups is low, which we have pointed out clearly in the discussion. However, we know that ART samples are not a homogenous group, and that the only way to understand the changes and the mechanisms associated with ART procedures, subfertility, and sex is to use subgroups. We think that the information provided by the subgroups is valuable in helping to outline the complex etiology of ART- and subfertility-associated phenotypes, and therefore an essential part of the work. Indeed, despite relatively small sample size in ICSI subgroup ($n = 21$) or even smaller in ICSI male subgroup ($n = 9$), we managed to identify interesting male infertility- and Y chromosome-associated genes (Rows 468-471: “Interestingly, the upregulated genes *ZFY*, *RPS4Y1*, *PCDH11Y*, and *DDX3Y* in the ICSI placentas locate on chromosome Y and the expressions of them all correlated significantly with each other in ART male placentas, but not in the controls (Supplementary Table S8).” And rows 539-542: “Despite of the small sample size in sex-specific analysis of ICSI samples, 16 DEGs were observed in ICSI male placentas ($n = 9$, control males: $n = 19$), including five upregulated genes, *DDX3Y*, *EIF1AY*, *ZFY*, *RPS4Y1*, and *PCDH11Y*, locating all on chromosome Y.”).

However, we noticed that the head circumference (cm) of ICSI female newborns was significantly smaller compared to control females ($n = 18$, $n = 35$, respectively) ($P = 0.018$), but not the P -value of the HC SD (z-score) ($P = 0.059$) as we told. Therefore we thought it was reasonable to remove the sentences about ICSI phenotypes from the result part (“According to the Finnish growth charts³⁸, the HC SD (z-score) of ICSI female newborns was significantly smaller compared to control females ($n = 18$, $n = 35$, respectively) ($P = 0.018$).”), as well as from the discussion: “and ICSI female newborns were smaller at birth – even significantly smaller HCs compared to controls were observed. Sex-specific differences in ICSI have been observed also in previous studies. A significantly higher number of trophoctoderm cells in ICSI male embryos was detected compared to their female counterparts, which was not observed when IVF male and female embryos were compared⁸⁰. Also previously, ICSI female newborns had significantly smaller BWs (SD) as well as HCs (SD) at the ages of 18 and 36 months (the information about the HC at birth was lacking) compared to controls⁸¹.”. Although this is an interesting finding, the sample size is small, and the clear significance of this observation was not reached.

1. In Table S4, not all probes are linked to a gene, however in Table S5 (DMRs), each DMR has a gene associated with it. Were only promoter DMRs reported, or was a different method used to link these regions to genes?

Neither all CpGs (Table S4) nor all DMRs (Table 5S) associated with a gene, although the higher number of associations between DMRs and genes can be observed. In DMR analyses, we have used DMRcate R package and all DMRs, not only DMRs in the promoter regions, have been reported.

2. It is fascinating that some of the top DMRs identified in blood are also in the placenta – e.g. CHRNE and NECAB3. This is only mentioned in the results. Can the authors elaborate a bit in discussion about what this might mean. Are we looking at a SNP effect across tissues (e.g. infertility associated)? Or is it a hormone effect that influences methylation in the oocyte or early embryo?

We agree, this is an interesting finding, and important as it validates our analyses. In addition to results part, we had mentioned only briefly in the discussion that “In addition to confirming previous ART-associated alterations observed in blood and placental samples, we found new candidate genes”. Now we have added to the first paragraph of the discussion: “Interestingly, some of the ART-associated DNAm changes in the placenta observed in the current study, have been reported previously in the EWASs of blood samples. These findings confirmed our analyses and may reflect early epigenetic effects of ART procedures, such as hormonal treatment, that remain in the cells’ epigenetic mitotic memory. However, the effects of subfertility or fertility-associated genetic variation on DNAm cannot be excluded in this study.”.

3. In Table S10, for STIX, the mean expression in IUI is 18.6, but the standard deviation (SD) is 2,017? The same large difference is seen in other genes in this table.

Thank you for this notice, we have now corrected these values.

4. In Table S11 and Table S13, where IVF v Controls and ICSI v Controls results are shown, the average for the IVF and ICSI groups should also be displayed in both tables. This will give the leader a sense of how different the effect of ICSI is on methylation.

Thank you for this comment. We have now added those values to the tables. Also, we added corresponding values to the tables of other group comparisons to provide more profound information from other comparisons as well.

5. In figure 4B, a bar graph with SD is used to represent the data. In the case of ICSI, this means that 4 data-points are presented by 3 data-points. It would be better and more transparent to show the data as dotplot showing the actual data.

We have now corrected Figure 4b.

Reviewer #2 (Remarks to the Author):

The paper provides a comprehensive analysis of genome-wide DNA methylation and gene expression in human placentas from pregnancies conceived through Assisted Reproductive Technology (ART). It explores the epigenetic and gene expression differences between ART and naturally conceived placentas, investigating the impact of various ART procedures and underlying subfertility. Key findings include ART-associated changes in placental development pathways, potential mechanisms underlying phenotypic features associated with ART, and insights into the molecular background of infertility. The study contributes valuable knowledge on how ART may influence placental function and development, highlighting the importance of considering ART procedures and subfertility effects in understanding the molecular alterations in placentas and their potential implications.

Strengths:

The study addresses a significant and under-explored area, providing valuable insights into the epigenetic and expression profiles of ART placentas. The methodology is robust, employing genome-wide analyses and including a comprehensive range of ART procedures and control groups.

Specific Comments:

1. For the acronyms and abbreviations: there are some missing terms that should be collected in the list of acronyms and abbreviations such as CpG.

We have now updated the list of abbreviations.

2. Introduction, line 95: the authors should briefly discuss evidence for the sentence “However, the results of EWASs are not consistent” and explain this.

Now we have modified this part of the introduction: “However, despite of some repetitive candidate loci, particularly related to imprinted genes, the ART-associated DNAm alterations are mainly inconsistent^{29, 30}. Furthermore, ART procedure-specific EWASs as well as genome-wide gene expression studies for placentas are scarce and studies controlled for placental cell type heterogeneity are lacking.”.

3. Introduction, line 99: the following three paragraphs are quite important for the whole manuscript since they are showing the key contributions and novelties of this study. For the audience to better understand the contributions of this study, the authors should write a topic sentence at the beginning of line 99 and summarize the major contributions before diving into the details.

We have modified this part of the introduction: “Here, to gain a deeper understanding of the ART- and subfertility-associated molecular changes and phenotypes, we explored ART-associated genome-wide DNAm by microarrays (Illumina’s Infinium MethylationEPIC) and gene expression by 3’ mRNA sequencing (mRNA-seq) of full-term placental samples from ART and naturally conceived singleton newborns (Table 1).”.

4. Introduction, line 111: a connection between the previous paragraph and this paragraph could help the introduction to be more cohesive.

Now we have modified the last paragraph of the introduction: “In addition to ART procedure-specific analyses, we performed sex-specific analyses by examining placentas of male and female newborns separately, including sex chromosomes.”.

5. Results Section 1: the wording in this paragraph seems to be not well-structured. It seems that the authors lumped all the facts and data presentation together and this makes the reviewer hard to find a logic chain in understanding this section. The reviewer suggests that the authors present the data in a more systematic and logical way: for example, by ART procedures or by different phenotypes. It is quite tough to understand this paragraph while hopping between different phenotypes together with between different ART procedures.

Thank you for this comment. We agree, this part of the results is challenging. Now we have modified the first paragraph and split it into two parts.

6. Results, line 315, line 325, line 336, line 342,: one particular issue related to the P-value/number of samples presentation is how they matched with different comparisons: which

P-value/case number matches with which group. This seems to be a consistent headache for the reviewer to understand the data throughout the results presentation. It would be better understandable if the authors could clearly define which P-value/case number is associated with which group throughout all results sections. Simply saying “respectively” did not make the understanding easier for the audience.

We have now modified sample sizes, *P*-values, and correlation coefficients according to the reviewer’s suggestion.

7. Results, line 359: the authors need to justify why choosing the DNAm difference of >5%.

Now we have modified the sentence about the DNAm difference of >5% in the results section: ”To separate the most prominent changes and to minimize false positive hits due to the observed inflation, we focused on the CpG sites with DNAm difference in the effect size of $\geq 5\%$ between ART and control placentas, which are termed as differentially methylated positions (DMPs).”.

8. Meanwhile, the reviewer supposes that the “difference” is equivalent to the “|delta_β| larger than 0.05”, and the “effect size larger than 0.05 or smaller than -0.05” in the Figures, right? If this is the case, the authors should clarify the usage of these three terms for the better understanding of the readers.

We have now corrected this, and the terms are consistent.

9. Results, line 370-371: the sentence needs to be rewritten to be clearer and grammatically correct.

This section has been rewritten after new DNAm analyses.

10. Results, line 389: how did the authors define “the most prominently downregulated”? Was that based on the significance level of effect size?

Thank you for this notice. Now we have changed it ”the most significantly”.

11. Results, line 391-402: the majority of the contents are more suitable to be placed in the discussion section.

We have condensed the text and placed part of it in the discussion section.

12. Results, line 411-412: this is an interesting finding and the authors should further discuss over this finding. Is this serendipity?

Since *DLK1* is a Notch ligand and *NOTCH3* is a Notch receptor, this is an interesting finding. There is a study, which suggests interaction between these two genes in the pigs’ skeletal muscle and cultured myocytes (Fu et al. *Animals*, 2022, doi: 10.3390/ani12121523) but clear evidence of direct interaction is still lacking, as we wrote. The functions of different Notch receptors (1-4) and their interaction with DLK1 are under intensive research and future studies will show if this finding is only a serendipity.

13. Results, line 434: the reviewer suggests that the authors rewrite this sentence and be more detailed.

Owing to the new DNAm results, we have modified the whole: “To separate the effects of *in vitro* culture from the effects of hormonal treatments and subfertility, we compared IUI, SF, and control placentas to the ART placentas. According to this comparison, only 37 DMPs linking to 26 genes and 47 DMRs linking to 53 genes were associated with *in vitro* culture in DNAm analysis (Fig. 1f, Supplementary Table S10). Interestingly, among these genes were *APC2*, *KIFC2*, *MUC5B*, *CELF5*, *KNDC1*, and *FAM83H-AS1*, which have previously been associated with IVF in the first-trimester placenta⁶². mRNA-seq analysis revealed 13 DEGs associating with *in vitro* culture (Fig. 1g, Supplementary Table S11).”.

14. Results, line 446-448: for better understanding, the authors should clearly separate the results from IVF and ICSI. It is quite confusing to understand the sentence that blend the results from both IVF and ICSI.

Now we have modified the paragraph.

15. Results, line 491: the transition is abrupt. There should be a logical connection between this paragraph and the previous paragraph. A sentence of motivation would suffice.

We have now modified this.

16. Line 781: please check spelling. Should it be “Epigenome” or “Eepigenome”?

We have now corrected this.

17. Table 1: the authors coded too much information using different colors in text and boxes, which makes the understanding of the table and table caption really hard. The table should be made for an easier way to understand.

Thank you for this comment. This table consists of all necessary information: all the analyses we have done, all the subgroups we have used for these analyses, and sex ratios. Also, information about the subgroups, which we did not use due to the low sample sizes is important and enable to gain comprehensive picture about the research material. However, owing to the DNAm analyses for IUI and SF, the current version of the Table I is easier to understand.

18. Fig. 1: the relationship between effect size and “delta_ β ” should be clarified in the figure caption.

We have now corrected this.

19. Fig. 2: The reviewer really likes the Venn plot. However, the reviewer is curious whether the overlaps are only calculated for DMPs, or did the authors also considered the positive/negative effect size? For example, are all the 5 DMPs overlapped in Fig. 2c with a positive effect size? Is it possible to incorporate the information of the “positive/negative” effect size when compared with the control group into the plot?

The definition for DMPs ($FDR < 0.05$, $\Delta\beta \leq -0.05$ and $\Delta\beta \geq 0.05$) has been indicated in materials and results sections, and now we have added it also to the figure texts (Figures 1, 2, and 3). According to this definition, there are both negative and positive effects sizes in the overlapping DMPs in the Venn diagrams.

20. Fig. 1 – Fig. 3: it seems that the authors used different thresholds for the effect size. Please justify the choices.

The definition for the DMPs is always the same ($FDR < 0.05$, $\Delta\beta \leq -0.05$ and $\Delta\beta \geq 0.05$), but due to the limited space in the Volcano plots, we named only the genes associated with the largest effect sizes. Now we have clarified the figure texts (Figures 1, 2, and 3): “All DMPs ($FDR < 0.05$, $\Delta\beta \leq -0.05$ and $\Delta\beta \geq 0.05$) are shown in green, and for visualization, the DMPs with the largest effect sizes ($\Delta\beta \leq -0.065/-0.055/-0.057$ and $\Delta\beta \geq 0.065/0.055/0.057$) are labeled based on UCSC RefGene Name.”.

Additional areas for improvement:

1. It would be beneficial to expand on the clinical implications of the observed changes, particularly how they might affect pregnancy outcomes or offspring health.

We agree, it is important to bring out potential clinical implications of the observed changes. We have brought out the potential connections between the observed changes and pregnancy outcomes and offspring health in many parts of the discussion (see below), and we prefer to be cautious about drawing further conclusions at this stage.

Rows 648-654: “Interestingly, it has been suggested that differences in placental mesenchymal stromal cells could lead to impaired vascular development and consequently to restricted growth^{80,81}. This is supported by our gene expression analysis, in which the most prominent ART-associated changes were linked to vasculogenesis, and where the downregulation of *NOTCH3* and *TRIM28*, both involved in the regulation of angiogenesis through the VEGFR-DLL4-NOTCH signaling circuit⁸², was observed.”.

Rows 679-686: “In human, mutations in *DLK1* gene have been reported as a cause of central precocious puberty associated with obesity and metabolic syndrome with undetectable *DLK1* serum levels⁸⁷. Furthermore, in Temple syndrome, where *DLK1* expression is downregulated due to the maternal uniparental disomy of the imprinted *DLK1* locus, phenotypic features such as prenatal growth failure, short postnatal stature, female early onset puberty, and truncal obesity have been observed⁸⁸. Since increased risks for SGA, rapid postnatal growth, and female early onset puberty⁶ have all been associated with ART children, *DLK1* is a plausible candidate gene for the phenotypic effects associated with ART.”.

Rows 730-734: “However, in the ART-associated phenotype the increased risk for LBW, SGA, rapid postnatal growth, and metabolic disorders would be a consequence of downregulation of *DLK1* caused by subfertility associated disorder, not a consequence of poor maternal nutrition as in the example of fetal programming. Notable, significant associations between subfertility and LBW have been observed in previous studies⁹⁸.”.

2. The study could further investigate/discuss the potential long-term effects of the identified epigenetic changes, exploring their persistence and impact beyond neonatal development.

Thank you for this comment. We have discussed about DOHaD, which concerns the long-term effects of prenatal environment (rows 708-734 in the discussion). Furthermore, owing to the comment of Reviewer #1, we have now added to the discussion section: “Interestingly, some of the ART-associated DNAm changes in the placenta observed in the current study, have been reported previously in the EWASs of blood samples. These findings confirmed our analyses and may reflect early epigenetic effects of ART procedures, such as hormonal treatment, that remain in the cells’ epigenetic mitotic memory. However, the effects of subfertility or fertility-associated genetic variation on DNAm cannot be excluded in this study.”.

3. While the study includes a diverse range of ART procedures, including more detailed information on the specific ART protocols used (e.g., hormone dosages, culture conditions) could enhance the understanding of procedure-specific effects.

Unfortunately, we were not able to take methodological details such as hormone dosages or culture conditions into account. These limitations of the study have been expressed in the discussion section: “We are aware of the limitations in this study. Owing to the limited sample size, we were not able to perform sex-specific analyses for all the subgroups, or focus on the details of the procedures, such as the length of embryo culture, hormonal treatments, or freezing methods.”.

Summary:

The paper makes a significant contribution to the field of reproductive medicine and epigenetics, offering a foundation for future research into the mechanisms by which ART may impact placental development and function. Further studies are encouraged to build on these findings, exploring the clinical relevance and potential interventions to mitigate adverse effects associated with ART.

Reviewer #3 (Remarks to the Author):

The manuscript by Auvinen et al. studied DNA methylation alterations and gene expression aberrations in placentas of ART pregnancies. The authors tried to additionally delineate the effect of various ART procedures and infertility on the placental methylome and transcriptome. This included a separate gender focused analysis to study epigenetic changes on sex chromosomes. Overall, the study is well designed and the findings will be of interest for researchers working on medically assisted reproduction. Furthermore, the manuscript is well written and the results are well described. One of the main limitations of this study is the low sample number, a point which the authors discussed in their manuscript.

1. The authors corrected for several covariates including cell type, gender, and maternal age and pre-pregnancy BMI. As evident from Supplementary Table 1, gestational age differs significantly between ART vs control pregnancies as well as IVF vs ICSI. Did the authors consider adjusting for gestational age in their model to account for this difference?

Thank you for this comment. The model was adjusted by batch, cell type, sex, maternal age, and BMI according to the singular value decomposition (SVD) plot. According to the plot, we did not consider the gestational age as a significant covariate.

2. The authors mentioned the following in their analysis focused on imprinted genes (Lines 593-596): “To focus specifically on ICRs, we compared 826 CpGs on ICRs compiled by Ochoa and colleagues between ART and control placentas and observed significant differences in 115 sites (90 hypo- and 25 hypermethylated) ($P < 0.05$) (Supplementary Table S32). After multiple testing correction, 16 CpG sites remained significant (Bonferroni adjusted $P < 0.05$).” Please indicate whether multiple testing correction was performed to correct for 826 measurements or for the entire set of CpG sites measured via EPIC arrays.

The multiple testing correction was performed for those 826 CpG sites on ICRs. Now we have modified the sentence: “After multiple testing correction for those 826 sites, 25 CpG sites remained significant (Bonferroni adjusted $P < 0.05$).”.

Minor:

a. In Lines 443-444, the authors mention: “Only five DMPs and one gene (TCN1) as well as 17 DMR-associated genes were common between the IVF and ICSI placentas and when the IVF placentas were compared to ICSI, none DMPs or DMRs were detected.” Please correct the typo in the sentence by replacing “none” with “no”.

We have now corrected this.

b. In the results section Lines 586-587, the following is included: “Imprinted genes are a group of parent-of-origin monoallelically expressed genes, which are maintained by correctly methylated ICRs established in the germline.” This definition of imprinted genes should be included in the Introduction or Discussion section with its reference and not in the Results section.

We have now modified this part of the manuscript and part of the sentence is in the introduction section: “Increasing evidence suggest that the placenta is particularly susceptible to epigenetic changes caused by ART and/or infertility²³⁻²⁵ and alterations have been found especially in epigenetically controlled repetitive elements (REs)²⁶⁻²⁸ and parent-of-origin monoallelically expressed imprinted genes¹⁹⁻²².” To focus specifically on ICRs, which are established in the germline to control imprinted gene expression, we compared 826 CpGs on ICRs compiled by Ochoa and colleagues⁵⁴ between ART and control placentas and observed significant differences in 148 sites (108 hypo- and 40 hypermethylated) ($P < 0.05$) (Supplementary Table S32).”.